# WorldSplat: Gaussian-Centric Feed-Forward 4D Scene Generation for Autonomous Driving

**Ziyue Zhu**[1,2], **Zhanqian Wu**[2], **Zhenxin Zhu**[2], **Lijun Zhou**[2], **Haiyang Sun**[2], **Bing Wang**[2], **Kun Ma**[2], **Guang Chen**[2], **Hangjun Ye**[2], **Jin Xie**[✉3], **Jian Yang**[✉1,3]

[1]PCA Lab, VCIP, College of Computer Science, Nankai University    [2]Xiaomi EV
[3]School of Intelligence Science and Technology, Nanjing University, Suzhou

zhuziyue@mail.nankai.edu.cn   csjxie@nju.edu.cn   csjyang@nankai.edu.cn

**Project Page:** https://wm-research.github.io/worldsplat/

## Abstract

Recent advances in driving-scene generation and reconstruction have demonstrated significant potential for enhancing autonomous driving systems by producing scalable and controllable training data. Existing generation methods primarily focus on synthesizing diverse and high-fidelity driving videos; however, due to limited 3D consistency and sparse viewpoint coverage, they struggle to support convenient and high-quality novel-view synthesis (NVS). Conversely, recent 3D/4D reconstruction approaches have significantly improved NVS for real-world driving scenes, yet inherently lack generative capabilities. To overcome this dilemma between scene generation and reconstruction, we propose WorldSplat, a novel feed-forward framework for 4D driving-scene generation. Our approach effectively generates consistent multi-track videos through two key steps: $(i)$ We introduce a 4D-aware latent diffusion model integrating multi-modal information to produce pixel-aligned 4D Gaussians in a feed-forward manner. $(ii)$ Subsequently, we refine the novel view videos rendered from these Gaussians using a enhanced video diffusion model. Extensive experiments conducted on benchmark datasets demonstrate that WorldSplat effectively generates high-fidelity, temporally and spatially consistent multi-track novel view driving videos.

## 1 Introduction

Synthesizing realistic driving-scene videos with controllable viewpoints is a key challenge in autonomous driving and computer vision, crucial for scalable training and closed-loop evaluation. Recent generative models (Mao et al., 2024; Gao et al., 2023; Wen et al., 2024) have advanced high-fidelity, user-defined video generation, reducing reliance on costly real data. Meanwhile, urban scene reconstruction methods (Chen et al., 2024c; Yan et al., 2024) have improved 3D representations (Mildenhall et al., 2021; Kerbl et al., 2023) and consistent novel-view synthesis.

Despite advancements, generation and reconstruction approaches face a dilemma between unseen-environment creation and novel view synthesis. Existing video generators (Mao et al., 2024; Gao et al., 2023; Wen et al., 2024; Li et al., 2024a; Gao et al., 2024b) work in the 2D image domain and often lack 3D consistency and novel-view controllability: they may look plausible from one angle but fail to stay coherent when generating from new viewpoints. Meanwhile, scene reconstruction methods (Yang et al., 2023; Yan et al., 2024; Chen et al., 2024c) achieve accurate 3D consistency and photorealistic novel views from recorded driving logs, yet they lack generative flexibility, being unable to imagine scenes beyond the captured data. Although video generation followed by reconstruction (Gao et al., 2024a; Lu et al., 2024; Mao et al., 2024) is feasible, the quality of the resulting novel views remains constrained by both processes. Thus, bridging generative imagination with faithful 4D reconstruction remains an open challenge.

---

✉ Corresponding authors: Jian Yang and Jin Xie.

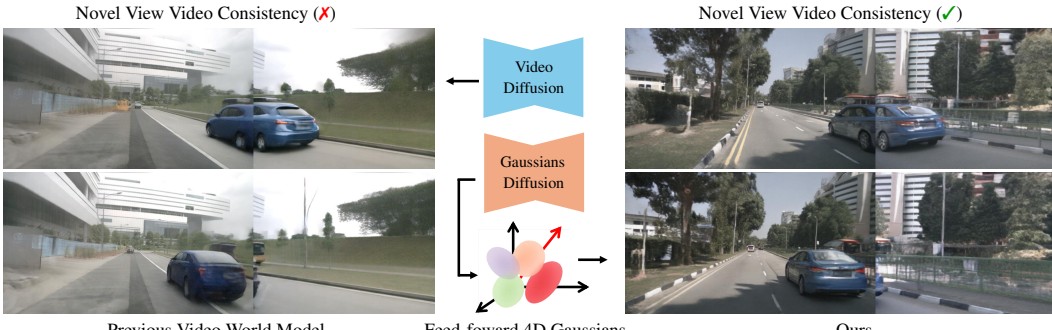

Figure 1: Comparison of different driving world models. Previous driving world models (Jiang et al., 2024; Gao et al., 2023) focus on video generation, while our method directly creates controllable 4D Gaussians in a feed-forward manner, enabling the production of novel-view videos (*e.g.* shifting ego trajectory $\pm Nm$) with spatiotemporal consistency.

To address these challenges, we introduce WorldSplat, a feed-forward framework that combines generative diffusion with explicit 3D reconstruction for 4D driving-scene synthesis. Our framework creates a dynamic 4D Gaussian representation and renders the novel views along any user-defined camera trajectory without per-scene optimization. By embedding 3D awareness into the diffusion model and using an explicit Gaussian-centric world representation, our method ensures spatial and temporal consistency across novel trajectory views. As shown in Fig. 1, prior driving world models (Gao et al., 2023; Mao et al., 2024; Jiang et al., 2024) produce realistic videos but often lose coherence when synthesizing novel-view sequences due to their stochastic nature. In contrast, WorldSplat directly outputs a 4D Gaussian field in a single forward pass, enabling stable, consistent novel-view rendering. For flexible generation control, the framework supports rich conditioning inputs—including road sketchs, textual descriptions, dynamic object placements, and ego trajectories—making it a highly controllable simulator for diverse driving scenarios.

Specifically, WorldSplat operates in three stages. First, without relying on real images (Ren et al., 2025) or LiDAR (Wang et al., 2024d), but only on flexible user-defined control conditions, a **4D-aware latent diffusion model** generates multi-view, temporally coherent latents inherently containing RGB, metric depth, and semantic channels, offering perspective-aware 3D information of both visual appearance and scene geometry. Next, a **latent Gaussian decoder** converts these latents into an explicit *4D Gaussian scene representation*. In particular, it predicts a set of pixel-aligned 3D Gaussians, which are separated into static background and dynamic objects and then aggregated into a unified 4D representation, explicitly modeling a dynamic driving world. For clarity, we visualize this representation in Appendix Fig. 7, and argue that it provides a more suitable basis than point maps (Wang et al., 2025a; Guo et al., 2025) for consistent novel-view video generation. We then apply fast Gaussian splatting to project the Gaussians into novel camera views, enabling *novel-track video synthesis* with true geometric consistency. Finally, an **enhanced diffusion refinement model** is applied to the rendered frames to further improve realism by correcting artifacts and enhancing fine details (e.g., texture and lighting), yielding high-fidelity novel-view videos. To summarize, our main contributions include:

- **Framework of 4D scene generation.** We introduce WorldSplat, a feed-forward 4D generative framework that unifies driving-scene video generation with explicit dynamic scene reconstruction.

- **Feed forward Gaussians decoder.** We propose a dynamic aware Gaussian decoder that directly infers precise pixel-aligned Gaussians from multimodal latents and aggregates them into a 4D Gaussian representation with static-dynamic decomposition.

- **Comprehensive evaluation.** Extensive experiments show that our framework generates spatially and temporally consistent free-viewpoint videos, achieves state-of-the-art performance in driving video generation, and provides significant benefits for downstream driving tasks.

## 2 RELATED WORK

**Driving World Models.** Recent world models for autonomous driving (Gao et al., 2023; 2024c; Li et al., 2024a; Swerdlow et al., 2024; Hu et al., 2023; Jiang et al., 2024; Li et al., 2024b; Wang et al., 2024f) have advanced the simulation of realistic street scenes, aiming to generate diverse synthetic data for robust driving systems. Most approaches rely on vision as the primary modality and focus on generating high-fidelity driving videos. For instance, GAIA-1 (Hu et al., 2023) synthesizes realistic driving scenarios, while DriveDreamer (Wang et al., 2024e) learns policies from real-world data. Vista (Gao et al., 2024c) scales to large driving datasets, and MagicDrive (Gao et al., 2023) ensures cross-camera consistency via attention mechanisms. Subsequent works (Swerdlow et al., 2024; Huang et al., 2024a; Chen et al., 2024a; Ma et al., 2024; Wen et al., 2024; Guo et al., 2024; Zeng et al., 2025) further improve controllability, video length, and visual quality.

Beyond video generation, recent studies explore 3D and 4D scene modeling (Li et al., 2024a; Lu et al., 2024; Mao et al., 2024; Zheng et al., 2024; Wang et al., 2024b; Ren et al., 2024; 2025; Wang et al., 2024a). MagicDrive3D (Gao et al., 2024a) supports multi-condition 3D scene generation, while InfiniCube (Lu et al., 2024) produces unbounded dynamic 3D scenes. DreamDrive (Mao et al., 2024) further extends to generalizable 4D generation. However, these approaches often rely on video-first pipelines, leading to reconstruction artifacts and sparse-view inconsistencies, underscoring the need for direct generation of coherent 3D/4D representations.

**Urban Scene Reconstruction.** Urban scene reconstruction and novel-view synthesis are commonly tackled with neural 3D representations (Mildenhall et al., 2021; Barron et al., 2021; Kerbl et al., 2023), but driving scenes remain difficult due to sparse views and dynamic objects. Gaussian-based methods (Yan et al., 2024; Zhou et al., 2024a;b; Chen et al., 2024c) use bounding boxes to reconstruct static and dynamic parts, while self-supervised methods (Yang et al., 2023; Huang et al., 2024b; Chen et al., 2023; Peng et al., 2024) decompose them automatically. Feed-forward approaches (Wei et al., 2024; Yang et al., 2024a; Wang et al., 2025b;a) further speed up reconstruction by avoiding per-scene optimization. However, these works focus on reconstruction rather than generation. Our method bridges this gap, enabling feed-forward 4D scene generation with high-quality novel views.

## 3 METHOD

### 3.1 OVERVIEW

As illustrated in Fig. 2, our framework comprises three key modules: a 4D-aware latent diffusion model (Sec. 3.2) for multi-modal latent generation, a latent Gaussian decoder (Sec. 3.3) for feed-forward 4D Gaussian prediction with real-time trajectory rendering, and an enhanced diffusion model (Sec. 3.4) for video quality refinement. The three modules are trained independently, and their architectures and training procedures are detailed in Secs. 3.2–3.4. Finally, Sec. 3.5 describes how these modules are integrated to generate high-fidelity, spatiotemporally consistent videos.

### 3.2 4D-AWARE DIFFUSION MODEL

Given a noise latent and fine-grained conditions (*i.e.*, bounding boxes, road sketch, captions and ego trajectory), the 4D-Aware Diffusion Model performs denoising to generate multi-modal latents encompassing RGB, depth, and dynamic object information, which are subsequently used for 4D Gaussian prediction. In the following, we elaborate on the latents, control conditions, model architecture, and training strategy.

**Multi-modal latent integration.** Given a $K$-view driving video clip with $T$ frames $\mathcal{I} = \{\mathbf{I}_t^v\}$, we first extract a multi-view image latent $\mathbf{L}_{\text{img}} = \mathcal{E}(\mathcal{I})$ using a pretrained VAE encoder (hpcai tech, 2024). Next, we generate metric depth maps $\mathcal{D}$ with a foundation depth estimator (Hu et al., 2024), normalize them to $[-1, 1]$, replicate across three channels, and encode them into a depth latent $\mathbf{L}_{\text{depth}}$. Prior works (Wei et al., 2024; Go et al., 2024; Yang et al., 2024b) have shown that incorporating depth latents improves both 3D reconstruction and generation quality. To further separate static and dynamic objects for 4D scene reconstruction, we obtain a semantic mask latent $\mathbf{L}_{\text{seg}}$ from binary segmentation masks of dynamic class objects using SegFormer (Xie et al., 2021). Finally, we concatenate the three latents channel-wise to form the decoder input $\mathbf{L} = concate\{\mathbf{L}_{\text{img}}, \mathbf{L}_{\text{depth}}, \mathbf{L}_{\text{seg}}\}$.

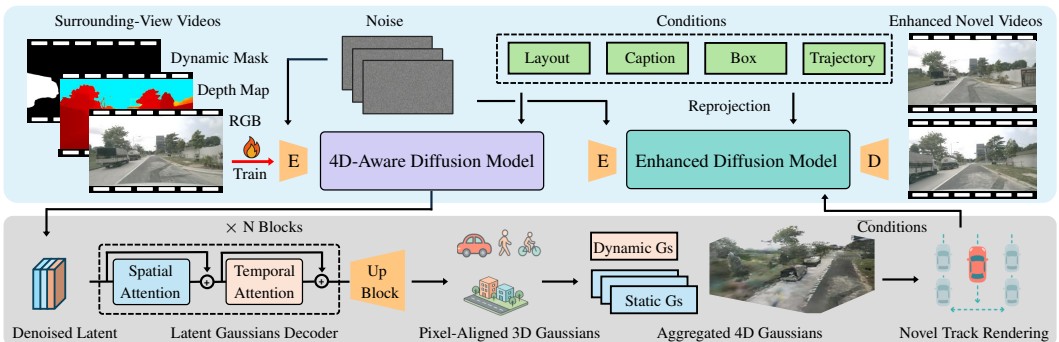

Figure 2: The overview of our framework. (1) Employing a 4D-aware diffusion model to generate a multi-modal latent containing RGB, depth, and dynamic information. (2) Predicting pixel-aligned 3D Gaussians from the denoised latent using our feed-forward latent decoder. (3) Aggregating the 3D Gaussians with dynamic-static decomposition to form 4D Gaussians and rendering novel-view videos. (4) Improving the spatial resolution and temporal consistency of the rendered videos with an enhanced diffusion model. "E" and "D" denote the VAE encoder and decoder, respectively. The ↑ arrow and the ↑ ones denote the train-only and inference.

**Multi-Conditions Control.** The diffusion transformer conditions on structured cues, including BEV layout $\mathcal{S}$, instance boxes $\mathcal{B}$, ego trajectory $\mathcal{T}$, and text descriptions $\mathcal{D}$, denoted collectively as $\mathcal{C} = \{\mathcal{S}, \mathcal{B}, \mathcal{T}, \mathcal{D}\}$. Following MagicDrive (Gao et al., 2023; 2025), we derive the layout, box, and trajectory inputs. For fine-grained caption control, we introduce *DataCrafter*, which segments a $K$-view video into clips, scores them with a VLM evaluator (Wang et al., 2024c), generates per-view captions, and fuses them via a consistency module. The resulting structured captions capture both scene context (weather, time, layout) and object details (category, box, description), ensuring temporal coherence and spatial consistency across views.

**Architecture.** The architectures of our 4D-Aware Diffusion models are ControlNet-based (Chen, 2023) transformers built upon OpenSora v1.2 (hpcai tech, 2024). Specifically, we extend OpenSora with a dual-branch Diffusion Transformer: a main DiT stream for spatiotemporal video latents $\mathbf{L}$ and a multi-block ControlNet branch for the conditions $\mathcal{C}$. To ensure multi-view coherence, we replace standard self-attention with cross-view attention.

Each ControlNet block integrates road sketch latents $\mathcal{E}(\mathcal{S})$ from a pretrained VAE (hpcai tech, 2024) and text embeddings from a T5 encoder (Raffel et al., 2020). The 3D boxes, ego trajectory, and text features are further fused through cross-attention into a unified scene-level signal, enabling fine-grained guidance and consistent video synthesis across time and viewpoints.

**Training.** In training stages, we replace the standard IDDPM scheduler with an Rectified Flow model to improve stability and reduce the number of inference steps. Let $\boldsymbol{x} \sim \boldsymbol{p}_{\text{data}}$ be a real sample of the clean latent $\mathbf{L}$ and $\boldsymbol{\epsilon} \sim \mathcal{N}(\mathbf{0}, \mathbf{I})$ be a noise sample. We introduce a continuous mixing parameter $s \in [0, 1]$ and define the interpolated state

$$\boldsymbol{z}(s) = (1-s)\boldsymbol{\epsilon} + s\boldsymbol{x}. \tag{1}$$

We then train a neural field $g_\psi(\boldsymbol{z}, s, \mathcal{C})$, conditioned on $\mathcal{C}$, to recover the target vector $\boldsymbol{x} - \boldsymbol{\epsilon}$ by minimizing

$$\mathcal{L}(\psi) = \mathbb{E}_{\boldsymbol{x},\boldsymbol{\epsilon},s}\left\|g_\psi(\boldsymbol{z}(s), s, \mathcal{C}) - (\boldsymbol{x} - \boldsymbol{\epsilon})\right\|_2^2. \tag{2}$$

At test time, we discretize $s_k = k/N$ for $k = N, \ldots, 1$ and step backward via

$$\boldsymbol{z}(s_{k-1}) = \boldsymbol{z}(s_k) - \frac{1}{N}g_\psi(\boldsymbol{z}(s_k), s_k, \mathcal{C}). \tag{3}$$

At inference, rather than using a VAE decoder to reconstruct video frames from denoised latents, we employ our Latent 4D Gaussian Decoder (Sec. 3.3) to directly predict the 4D Gaussians for novel view videos rendering.

### 3.3 LATENT 4D GAUSSIANS DECODER

Our Gaussians Decoder predicts pixel-aligned 3D Gaussians from the multi-modal latents $\mathbf{L}$ (Sec. 3.2). We then leverage the semantic information within the latents to distinguish dynamic from static objects and reconstruct the 4D scene from the 3D Gaussians.

**Architecture.** Our transformer-based decoder (Dosovitskiy et al., 2020; Yang et al., 2024a; Zhang et al., 2024) consists of multiple cross-view attention blocks and temporal attention layers across frames, followed by a hierarchy of up-sampling blocks to predict per-pixel Gaussian parameters. As shown in Fig. 2, this design captures the spatio-temporal dynamics of 4D scenes and directly outputs pixel-aligned 3D Gaussians from the multi-modal latent input $\mathbf{L}$. To further enhance 3D spatial cues, we incorporate the Plücker (Plucker, 1865) ray map $\mathbf{P}$, which encodes pixel-wise ray origins $\mathbf{R}_o$ and directions $\mathbf{R}_d$ derived from camera intrinsics and extrinsics.

Each 3D Gaussian (Kerbl et al., 2023) is parameterized as $\boldsymbol{g} = (\boldsymbol{\mu}, \boldsymbol{r}, \boldsymbol{s}, \boldsymbol{\alpha}, \boldsymbol{c})$, where $\boldsymbol{\mu} \in \mathbb{R}^3$, $\boldsymbol{r} \in \mathbb{R}^4$, $\boldsymbol{s} \in \mathbb{R}^3$, $\boldsymbol{\alpha} \in \mathbb{R}^+$, and $\boldsymbol{c} \in \mathbb{R}^3$ denote center, quaternion rotation, scale, opacity, and color, respectively. The final decoder layer predicts per-pixel offsets $\boldsymbol{\delta}$, rotation $\boldsymbol{r}$, scale $\boldsymbol{s}$, opacity $\boldsymbol{\alpha}$, color $\boldsymbol{c}$, depth $\boldsymbol{d}$, and logits $\boldsymbol{m}$ for static–dynamic classification. The Gaussian center is then computed as $\boldsymbol{\mu} = \mathbf{R}_o + \boldsymbol{d} \odot \mathbf{R}_d + \boldsymbol{\delta}$, where $\boldsymbol{\delta} \in \mathbb{R}^3$ is the learned offset. This process yields a sequence of Gaussian sets $\mathcal{G}$ and mask $\mathcal{M}$ indicating dynamic class objects over time, which can be compactly written as:

$$D_\phi : (\mathbf{L}_{\text{img}}, \mathbf{L}_{\text{depth}}, \mathbf{L}_{\text{seg}}, \mathbf{P}) \ \mapsto \ \{(\mathbf{G}_t, \mathbf{M}_t) \in \mathbb{R}^{V \times H \times W \times (14,1)}\}_{t=1}^T. \tag{4}$$

Compared to prior feed-forward scene reconstruction models (Wei et al., 2024; Yang et al., 2024a; Charatan et al., 2024; Chen et al., 2024b), our decoder supports over 48 simultaneous input views, enabling a more comprehensive reconstruction of complex scenes.

**4D Gaussians Aggregation.** By merging the 3D Gaussian estimates from each frame, we form an scene model that remains temporally aligned and coherent. We adopt a straightforward 4D reconstruction scheme: Given the known ego trajectory $\mathcal{T}$, all 3D Gaussians are transformed into a unified coordinate system with ego-coordinate transformations. At each time step, we fuse the static Gaussians gathered from every frame with the dynamic Gaussians extracted from the current frame.

$$\mathcal{G}_{4D} \ = \ \Big\{ (\mathbf{G}_t \odot \mathbf{M}_t) \ \cup \ \bigcup_{i=1}^T \big(\mathbf{G}_i \odot (1 - \mathbf{M}_i)\big) \Big\}_{t=1}^T. \tag{5}$$

By integrating data from multiple time steps, our decoder captures the scene's complete geometry, appearance, and motion, enabling rendering from both new spatial viewpoints and different moments.

**Supervision and Loss functions.** Note that our Gaussian Decoder predicts both pixel-aligned 3D Gaussians and semantic masks to distinguish dynamic from static regions. The predicted semantic masks are supervised by those generated from SegFormer (Xie et al., 2021) using a binary cross-entropy loss. After assembling the 4D Gaussians with predicted masks across all observed timesteps, we project them onto a set of target rendering timesteps. During training, we randomly select a base timestep $t$ and sample $T$ target timesteps $\{t_i\}_{i=1}^T$ and extract the corresponding clean Latent $\mathbf{L}$ as input. For each $t_i$, we render RGB images $\mathcal{R}$ and depth images and supervise them with the corresponding ground-truth signals: RGB inputs $\mathcal{I}$ and metric depth maps (Hu et al., 2024). RGB reconstruction is guided by a combination of photometric $L_1$ loss and perceptual LPIPS loss (Zhang et al., 2018), while depth predictions are supervised with an $L_1$ loss in metric space. The overall training objective is defined as a weighted sum of these losses:

$$\mathcal{L} = \mathcal{L}_{\text{recon}} + \lambda_1 \, \mathcal{L}_{\text{lpips}} + \lambda_2 \, \mathcal{L}_{\text{depth}} + \lambda_3 \, \mathcal{L}_{\text{seg}}. \tag{6}$$

At inference, the 4D Gaussians generated by our pretrained Gaussian Decoder are used to render novel-view videos $\mathcal{R}'$ following customized ego trajectories $\mathcal{T}'$.

### 3.4 ENHANCED DIFFUSION MODEL

The Enhanced Diffusion Model refines the RGB videos rendered from the 4D Gaussians, with the generation process conditioned on both the original inputs $\mathcal{C}$ and the rendered videos. This refinement

enriches spatial details and enforces temporal coherence, yielding the final high-fidelity novel-view sequences.

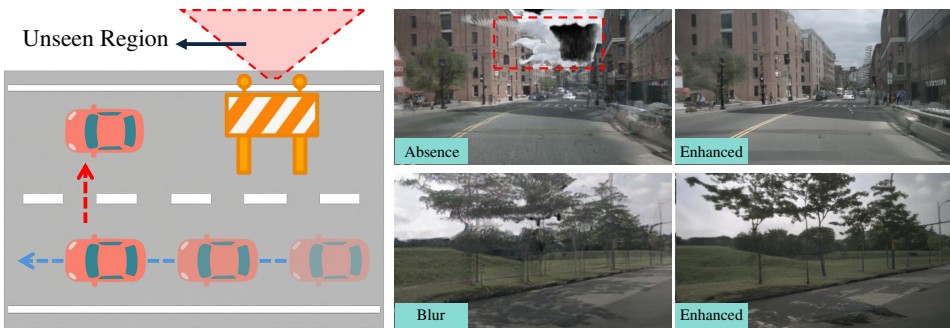

Figure 3: Effectiveness of the enhanced diffusion model. During novel-view video synthesis, rendering quality may degrade due to unobserved regions or high ego-vehicle speed, resulting in missing content and artifacts. Our enhanced diffusion model can inpaint unobserved areas and sharpen fast-motion frames.

**Reconstruction via Restoration.** As introduced in Sec. 3.3, our Latent Gaussians Decoder reconstructs 4D scenes in a feed-forward manner. However, inherent limitations of Gaussian splatting result in low-quality renderings for unobserved regions. Additionally, without per-scene optimization, novel-view reconstructions can become blurred under strong ego motion. To address these issues, we design an enhanced diffusion model to improve the quality.

As illustrated in Fig. 3 for better understanding, the upper-left novel-view rendering omits sky regions due to occlusions, and the lower-left rendering appears blurred due to strong ego motion. In contrast, our model substantially enhances the fidelity and clarity of the final renderings.

**Architecture and Training.** Its overall architecture and training strategy remain consistent with the 4D-Aware Diffusion Model Sec. 3.2. The objective is to refine the rendering results $\mathcal{R}$ in the latent space, with the ground truth being $\mathcal{E}(\mathcal{I})$. Thus, the regression target is the image latent $\mathcal{E}(\mathcal{I})$, consistent with common latent diffusion models (Jiang et al., 2024; Gao et al., 2025). The training pipeline mirrors that of the 4D-Aware Diffusion Model, differing only in the control conditions $\mathcal{C}' = \{\mathcal{R}, \mathcal{S}, \mathcal{B}, \mathcal{T}, \mathcal{D}\}$ and the regression target.

Due to the limitations of Gaussian splatting, novel-view renderings at inference often appear inferior to the source views used for training. ReconDreamer (Ni et al., 2024) reduces this gap by training with degraded renderings, but relying solely on degraded inputs weakens alignment between conditions and outputs. We instead adopt a mixed-conditioning strategy, combining degraded and high-quality views to improve both controllability and generation fidelity.

### 3.5 FRAMEWORK INFERENCE PIPELINE

During inference, the 4D-Aware Diffusion Model takes noise latents with control conditions $\mathcal{C}$ and outputs the denoised latent $\mathbf{L}_d$. The Gs decoder then predicts 4D Gaussians from $\mathbf{L}_d$, which are rendered into novel-view videos $\mathcal{R}'$ based on a customized ego trajectory $\mathcal{T}'$. Sketches and boxes are reprojected as $\mathcal{S}'$ and $\mathcal{B}'$, forming new control conditions $\mathcal{C}' = \{\mathcal{R}', \mathcal{S}', \mathcal{B}', \mathcal{T}', \mathcal{D}\}$. Taking noise latent and conditons $\mathcal{C}'$ as input, the Enhanced Diffusion Model refines $\mathcal{R}'$, producing high-quality novel-view videos.

**Customized Trajectory Selection.** Building on the ego-pose perturbation strategy of FreeVS (Wang et al., 2024d), we generate a set of novel tracks by laterally shifting the vehicle's path. Specifically, given the original ego-trajectory $\{\mathcal{T}_i\}_{i=1}^N$, we apply offsets $\Delta y \in \{\pm 1\,\text{m},\ \pm 2\,\text{m},\ \pm 4\,\text{m}\}$ along the vehicle's $y$-axis to produce six perturbed trajectories $\{\mathcal{T}_i + (0, \Delta y, 0)\}_{i=1}^N$. For each perturbed path, our aggregated 4D Gaussians render high-quality novel-view videos.

Furthermore, when reconstructing a scene from real driving videos, the 4D-Aware Diffusion Model is bypassed, and the Gs Decoder directly takes the clean latent as input.

# 4 EXPERIMENTS

## 4.1 EXPERIMENTAL SETUPS

**Dataset and Metrics.** We conduct experiments on the nuScenes benchmark (Caesar et al., 2020), which contains 1,000 urban driving scenes annotated at 2 Hz. We upsample the annotations (e.g., bounding boxes and road sketches) to 12 Hz following (Wang et al., 2023). The model is trained on 700 scenes and validated on 150. We evaluate generation quality using Fréchet Video Distance (FVD) (Unterthiner et al., 2019) and Fréchet Inception Distance (FID). For downstream evaluation, we measure the domain gap on perception tasks and assess how generated data improves perception model training.

**Implementation Details.** We adopt the pretrained OpenSora-VAE-1.2 (hpcai tech, 2024) as the backbone, fine-tuning only the cross-view attention blocks (Gao et al., 2023) in the diffusion transformer. More architectural and training details are provided in Sec. A.

## 4.2 ORIGINAL VIEW VIDEO GENERATION

**Quantitative Comparison.** In Tab. 1, we report the quantitative results of our video synthesis approach under three different conditioning schemes: (i) no first-frame guidance, (ii) first-frame guidance, and (iii) noisy latent initialization. Across all scenarios, our method consistently delivers the best scores on both the FVD and FID metrics.

Without first-frame guidance, our model achieves 74.13 $FVD_{multi}$ and 8.78 $FID_{multi}$, surpassing DriveDreamer-2 (Zhao et al., 2024), MagicDrive-V2 (Gao et al., 2025), and Panacea (Wen et al., 2024). Incorporating the first frame further boosts performance to 16.57 FVD and 4.14 FID, on par with or better than DriveDreamer-2 while maintaining temporal smoothness and structural detail. Under the noisy-latent protocol (6,019 clips), our method reaches 60.87 FVD and 6.51 FID, establishing a new state of the art over UniScene (Li et al., 2024a).

Table 1: Video generation comparison on the nuScenes (Caesar et al., 2020) validation set, with green and blue highlighting the best and second-best values, respectively.

| Method | Gen. Mode | Backbone | Video | Novel View | Sample Num | $FVD_{multi} \downarrow$ | $FID_{multi} \downarrow$ |
|---|---|---|---|---|---|---|---|
| DriveDreamer-2 | w/o first cond | SD-based | ✓ | ✗ | – | 105.10 | 25.00 |
| MagicDrive-V2 | w/o first cond | CogVideoX | ✓ | ✗ | – | 94.84 | 20.91 |
| MagicDrive3D | w/o first cond | SD-v1.5 | ✓ | ✓ | – | 164.72 | 20.67 |
| Panacea | w/o first cond | SD-based | ✓ | ✗ | – | 139.00 | 16.96 |
| DIVE | w/o first cond | OpenSora-v1.2 | ✓ | ✗ | 5369 | 94.60 | – |
| **Ours** | w/o first cond | OpenSora-v1.2 | ✓ | ✓ | 5369 | 74.13 | 8.78 |
| CoGen | w first cond | OpenSora-v1.2 | ✓ | ✗ | 5369 | 68.43 | 10.15 |
| DriveDreamer-2 | w first cond | SD-based | ✓ | ✗ | – | 55.70 | 11.20 |
| **Ours** | w first cond | OpenSora-v1.2 | ✓ | ✓ | 5369 | 16.57 | 4.14 |
| Vista* | w noisy latent | SVD | ✓ | ✗ | 6019 | 112.65 | 13.97 |
| UniScene | w noisy latent | DiT-based | ✓ | ✗ | 6019 | 70.52 | 6.12 |
| **Ours** | w noisy latent | OpenSora-v1.2 | ✓ | ✓ | 6019 | 60.84 | 6.51 |

**Qualitative Comparison.** In Fig. 4, we compare our generated videos with two leading methods: MagicDrive (Gao et al., 2023) and Panacea (Wen et al., 2024). We also show the real samples and the control inputs. As the results demonstrate, our approach produces more accurate shapes and positions for dynamic objects, and achieves much better consistency across multiple views. Overall, our method captures rich details with high realism while maintaining strong agreement between different viewpoints.

## 4.3 NOVEL VIEW SYNTHESIS

Following FreeVS (Wang et al., 2024d), we evaluate our method on novel trajectories using the FID and FVD metrics. Specifically, we translate the camera by offsets of $\pm 1$ m, $\pm 2$ m, and $\pm 4$ m, then compute FID and FVD between the generated RGB frames along each shifted trajectory and the original ground-truth frames.

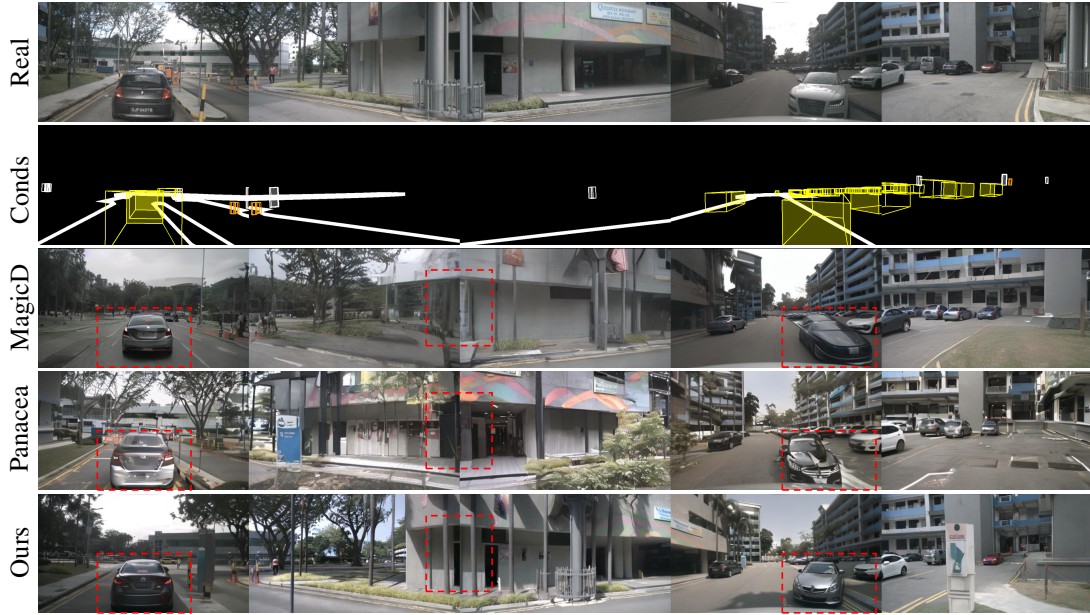

Figure 4: Comparison with MagicDrive (Gao et al., 2023) and Panacea (Wen et al., 2024). The top row shows real frames, the second row the corresponding sketches and bounding-box controls. Red boxes highlight areas where our method achieves the most notable improvements.

**Quantitative Comparison.** In Tab. 2, we compare WorldSplat with six baselines on nuScenes under viewpoint shifts of $\pm 1$, $\pm 2$, and $\pm 4$ meters. WorldSplat consistently achieves the best FID/FVD across all shifts—for example, at $\pm 1$ m it outperforms DiST-4D and OmniRe, and even at $\pm 4$ m it remains clearly ahead of all baselines. These results demonstrate the robustness and fidelity of our 4D Gaussian representation for novel-view synthesis under varying viewpoint shifts.

Table 2: Quantitative results of novel-view synthesis, reporting FID and FVD under viewpoint shifts of $\pm 1$, $\pm 2$, and $\pm 4$ meters. Baseline metrics are taken from DiST-4D (Guo et al., 2025).

| Method | Shift $\pm 1$m | | Shift $\pm 2$m | | Shift $\pm 4$m | |
| --- | --- | --- | --- | --- | --- | --- |
| | FID $\downarrow$ | FVD $\downarrow$ | FID $\downarrow$ | FVD $\downarrow$ | FID $\downarrow$ | FVD $\downarrow$ |
| PVG (Chen et al., 2023) | 48.15 | 246.74 | 60.44 | 356.23 | 84.50 | 501.16 |
| EmerNeRF (Yang et al., 2023) | 37.57 | 171.47 | 52.03 | 294.55 | 76.11 | 497.85 |
| StreetGaussian (Yan et al., 2024) | 32.12 | 153.45 | 43.24 | 256.91 | 67.44 | 429.98 |
| OmniRe (Chen et al., 2024c) | 31.48 | 152.01 | 43.31 | 254.52 | 67.36 | 428.20 |
| FreeVS[*] (Wang et al., 2024d) | 51.26 | 431.99 | 62.04 | 497.37 | 77.14 | 556.14 |
| DiST-4D (Guo et al., 2025) | 10.12 | 45.14 | 12.97 | 68.80 | 17.57 | 105.29 |
| **Ours** | 8.25 | 40.17 | 11.26 | 47.41 | 13.38 | 64.07 |

**Qualitative Comparison.** Furthermore, Fig. 5 offers a quantitative comparison with the state-of-the-art urban reconstruction model (Chen et al., 2024c), demonstrating that our method delivers superior spatial consistency. Our renderings are sharper and more detailed: OmniRe often loses fine elements such as lane markings and railings, whereas our approach preserves these features accurately. Moreover, our background reconstruction shows significant improvements over OmniRe.

## 4.4 ABLATION STUDY

In Tab. 3, we report FID and FVD for novel-view synthesis with a $\pm 2$ m ego shift across six variants to systematically validate each component's contribution.

We systematically validate each component's contribution: (1) **3D Gaussians Representation (Version A $\rightarrow$ B)**: Introducing 3D Gaussians as explicit scene representation significantly improves performance with FVD dropping from 260.07 to 75.26 ($-184.81$) and FID from 41.40 to 16.31

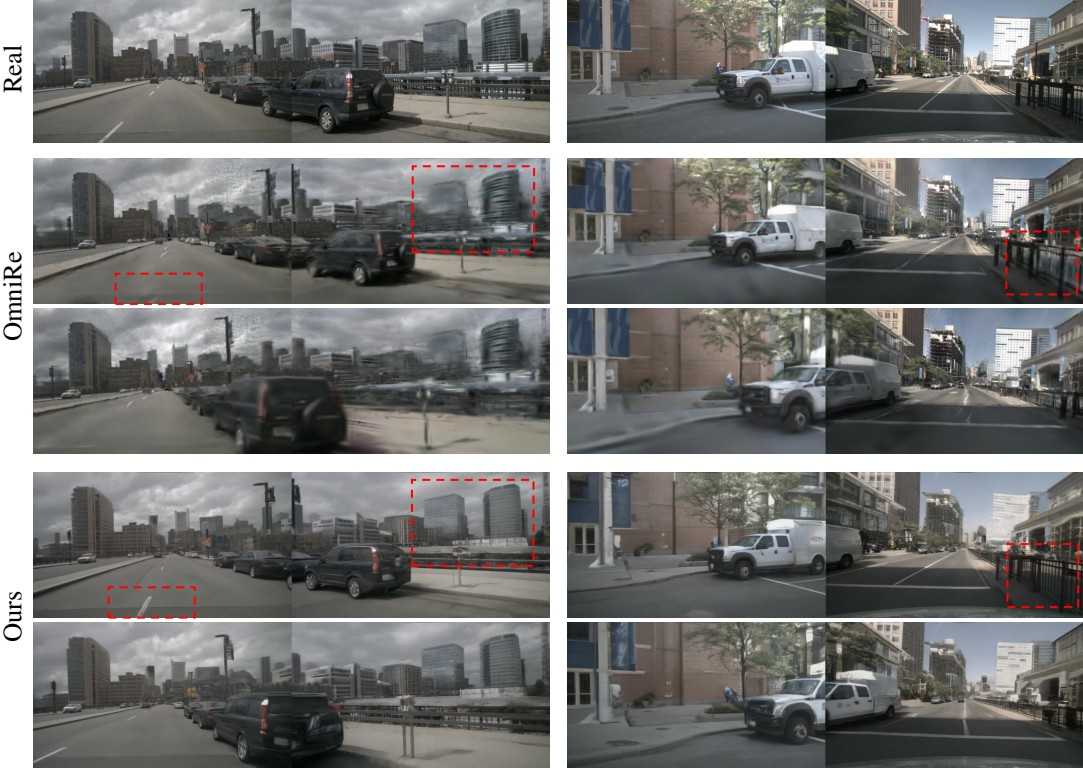

Figure 5: Qualitative comparison of our novel view synthesis against the state-of-the-art urban reconstruction method (Chen et al., 2024c). We translate the ego-vehicle by $\pm 2\,$m to generate the novel viewpoints. Red boxes indicate where our method achieves the greatest improvements.

Table 3: Ablation study of novel-view generation: "C-Reprojection" reprojects boxes and sketches; "3D Gs" uses Gaussians from single-frame reconstructions; "4D Gs" uses Gaussians from multi-frame reconstructions; "Mixed Aug" mixes renderings of varying quality during training; "Enhanced" employs the enhanced diffusion model to improve video quality.

| Version | C-Reprojection | 3D Gs | 4D Gs | Mixed Aug | Enhanced | FVD $\pm 2$m $\downarrow$ | FID $\pm 2$m $\downarrow$ |
|---------|:--------------:|:-----:|:-----:|:---------:|:--------:|:--------:|:--------:|
| A | ✓ | | | | ✓ | 260.07 | 41.40 |
| B | ✓ | ✓ | | | ✓ | 75.26 | 16.31 |
| C | ✓ | | ✓ | | ✓ | 50.73 | 11.60 |
| D | ✓ | | ✓ | ✓ | | 107.58 | 26.73 |
| E | | | ✓ | ✓ | ✓ | 51.64 | 12.07 |
| F | ✓ | | ✓ | ✓ | ✓ | 47.41 | 11.26 |

($-25.09$), demonstrating that 3D Gaussians enable spatiotemporally consistent videos with effective novel-view synthesis capability. (2) **4D Gaussian Aggregation (Version B $\rightarrow$ C)**: Aggregating single-frame 3D Gaussians into a unified 4D representation further improves performance with FVD from 75.26 to 50.73 ($-24.53$) and FID from 16.31 to 11.60 ($-4.71$). Single-frame Gaussians are often too sparse, producing holes and aliasing in novel views, while 4D aggregation enhances spatiotemporal consistency by densifying the representation across time in dynamic scenes. (3) **Mixed Augmentation (Version C $\rightarrow$ F)**: This contributes FVD improvement from 50.73 to 47.41 ($-3.32$) and FID from 11.60 to 11.26 ($-0.34$). As discussed in Line 292 of our paper, novel-view renderings at inference often appear inferior to source views; by degrading training source view quality through mixed augmentation, we improve the enhanced diffusion model's robustness and overall generation quality. (4) **Enhanced Diffusion (Version D $\rightarrow$ F)**: The Enhanced Diffusion module provides substantial improvements with FVD dropping dramatically from 107.58 to 47.41 ($-60.17$) and FID from 26.73 to 11.26 ($-15.47$). As detailed in Section 3.4 and illustrated in Figure 3, this module addresses inherent limitations of Gaussian splatting—low-quality renderings in unobserved regions

and blurred reconstructions under strong ego motion without per-scene optimization—by effectively inpainting missing content and sharpening fast-motion frames. (5) **Condition Reprojection (Version E → F)**: Removing reprojection conditions leads to 8.9% increase in FVD ($47.41 \rightarrow 51.64$) and 7.2% increase in FID ($11.26 \rightarrow 12.07$). While the rendering views from our 4D Gaussians already provide strong conditions, the condition reprojection (Section 3.5) further enhances performance by enforcing explicit 3D geometric constraints across views and frames.

### 4.5 DOWNSTREAM EVALUTAION

Beyond visual fidelity, we assess the domain gap on downstream tasks—3D detection and BEV map segmentation—following MagicDrive (Gao et al., 2023) (Tab. 4a). With a pretrained BEVFormer (Li et al., 2024c), our generated inputs achieve 38.49% mIoU and 29.32% mAP, outperforming DiVE by 2.53% and 4.79%.

Further, following the experimental setup of Panacea (Wen et al., 2024), we generate a new training dataset based on nuScenes and integrate the generated data with real data to train the StreamPETR (Wang et al., 2023) model. (Tab. 4b) reports the 3D object detection results, showing that our approach provides larger improvements over the baseline compared to Panacea.

Table 4: The applications of our method on the downstream tasks.

(a) Domain gap validation of generated data on driving perception with pretrained BEVFormer.

| Method | mIoU↑ | mAP↑ |
|---|---|---|
| MagicDrive (Gao et al., 2023) | 18.34 | 11.86 |
| MagicDrive3D (Gao et al., 2024a) | 18.27 | 12.05 |
| MagicDrive-V2 (Gao et al., 2025) | 20.40 | 18.17 |
| DiVE (Jiang et al., 2024) | 35.96 | 24.55 |
| Ours | 38.49 | 29.34 |
| Real | 44.09 | 34.51 |

(b) Performance gains achieved by incorporating generated data into the training of the StreamPETR.

| Method | mAP↑ | NDS↑ |
|---|---|---|
| Real | 34.5 | 46.9 |
| Panacea (Wen et al., 2024) | 22.5 | 36.1 |
| Real + Panacea (Wen et al., 2024) | 37.1 (+2.6) | 49.2 (+2.3) |
| Ours | 29.2 | 41.7 |
| Real + Ours | 38.5 (+4.0) | 50.1 (+3.2) |

## 5 FAILURE CASE STUDY

Our framework can fail when the customized ego trajectory enters completely unobserved regions (e.g., shifting a trajectory inside a building whose interior is never visible from original viewpoints). Without geometric priors from Gaussians' rendering, the enhanced diffusion model produces artifacts or blank regions. This limitation is inherent to view synthesis from limited observations and affects all reconstruction-based methods; future work could incorporate geometric inpainting or learned priors for heavily occluded regions.

## 6 CONCLUSION

In this work, we presented WorldSplat, a novel feed-forward framework that unifies the strengths of generative and reconstructive approaches for 4D driving-scene synthesis. By integrating a 4D-aware latent diffusion model with an enhanced diffusion network, our method produces explicit 4D Gaussians and refines them into high-fidelity, temporally and spatially consistent multi-track driving videos. Extensive experiments on standard benchmarks confirm that WorldSplat outperforms prior generation and reconstruction techniques in both realism and novel-view quality.

**Acknowledgement** This work was supported by the National Key R&D Program of China No. 2024YFC3015801, National Science Fund of China under Grant Nos. 62361166670, U24A20330, and 62276144.

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

# Appendix

## A  IMPLEMENTATION DETAILS

### A.1  ARCHITECTURES

In Fig. 6, we provide a detailed view of our enhanced diffusion model (Sec. 3.4). To enable fine-grained control over video synthesis, we condition on multiple signals: rendered RGBs from 4D Gaussians (Sec. 3.3), road sketches, 3D bounding boxes, ego-vehicle trajectories, and textual scene descriptions. The overall transformer backbone of our enhanced diffusion model is identical to that of our 4D-aware diffusion framework (Sec. 3.2); we simply adjust the input and output channel dimensions to suit different latent representations.

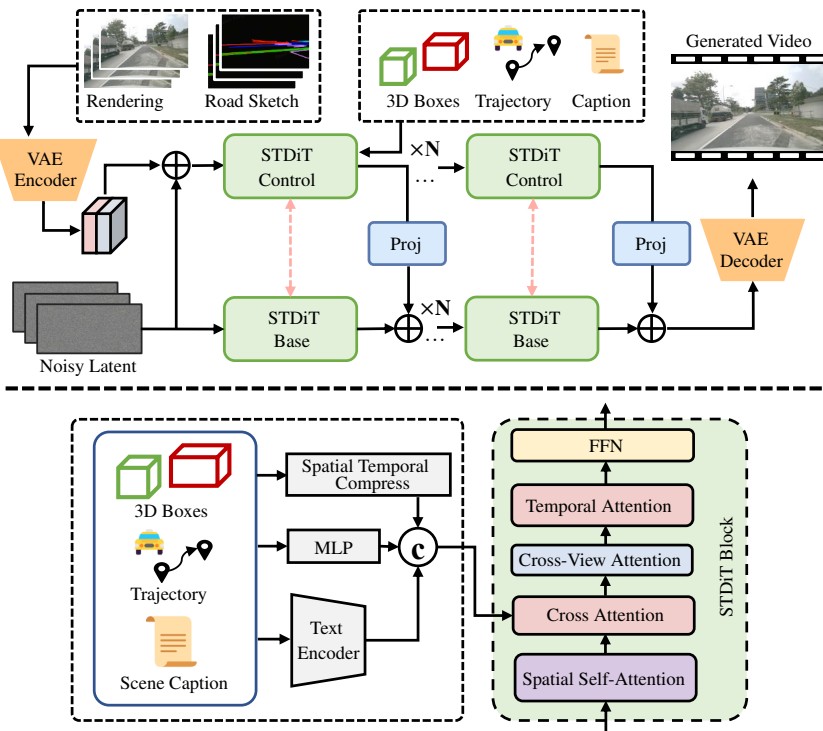

Figure 6: The architecture details of our diffusion transformer.

**Double-Branch Diffusion Transformer.** Following DiVE (Jiang et al., 2024), we first employ a frozen variational autoencoder (VAE) to encode the input multi-view video clip into a compact latent tensor $z \in \mathbb{R}^{V \times T \times C \times H \times W}$, where $V$ is the number of camera views, $T$ the number of frames, and $H, W$ the spatial dimensions of each latent feature map. A 3D patch embedding module then aggregates these features to capture spatiotemporal correlations. In parallel, we introduce a dedicated ControlNet (Chen, 2023) branch to inject rendering and sketch guidance: the VAE encodes both signals into latent patches, which are aligned with the main 3D patch embedder. We interleave specialized ControlNet blocks alongside each DiT transformer stage, merging sketch information into the main feature stream to achieve precise structural control.

**Spatial-Temporal Diffusion Transformer Block.** To enforce coherence across views without increasing parameter count, we replace standard self-attention with a cross-view attention mechanism. Concretely, given an input of shape $B \times V \times T \times H \times W \times C$, we reshape it to $B \times T \times (VHW) \times C$, treating the flattened $VHW$ dimension as the attention sequence length. This simple reordering enables cross-view interactions while keeping model size unchanged.

We further fuse 3D bounding boxes, ego-trajectory data, and scene captions via a single cross-attention layer. We project the 2D image-plane embeddings of each 3D box with a 3D convolution, encode the

ego trajectory through a small MLP, and tokenize the textual caption using a T5 backbone (Raffel et al., 2020). These modality-specific embeddings are concatenated and passed through a final MLP to produce a unified conditioning vector for the cross-attention block.

## A.2 TRAINING DETAILS

The training is organized into four sequential stages on $32\times$ H20 GPUs:

Table 5: 4D-Aware Diffusion Training Stages. Steps are reported for 8-GPU training. For 32-GPU training, divide by 4 (e.g., Stage 1: 15K steps).

| Stage | Description | Steps* | Resolution (6 views) | Time |
|---|---|---|---|---|
| **Stage 1** | Fine-tune from OpenSora v1.2 with layout/sketch control | 60K | 256×256 | ∼32h |
| **Stage 2** | Mixed-resolution training with varying frame lengths | 40K | 144p–360p | ∼28h |
| **Stage 3** | Switch to rectified flow at low resolution | 20K | 144p–360p | ∼14h |
| **Stage 4** | High-resolution fine-tuning with rectified flow | 60K | 480p–full | ∼83h |
| **Total** | **4D-Aware Diffusion Training** | **180K** | - | **∼157h** |

**Stage 1:** Starting from the OpenSora v1.2 checkpoints, we fine-tune for 60K iterations (15K steps on 32 GPUs) on 256×256 fixed-resolution images to establish layout and sketch control. At this stage, the ControlNet-Transformer, spatial attention, and layout module (with spatial self-attention in the base layers) are optimized.

**Stage 2:** We continue for 40K iterations (10K steps on 32 GPUs) using mixed resolutions (144p, 240p, 360p) and varying frame lengths, aligning the model to the nuScenes data distribution, still employing spatial self-attention.

**Stage 3:** DDPM is replaced with rectified flow. We first train for 20K iterations (5K steps on 32 GPUs) at low resolutions (144p–360p).

**Stage 4:** We finetune the model by 60K iterations (15K steps on 32 GPUs) at higher resolutions (480p to full scale) with rectified flow.

**Additional Training Stages:**

- **Gaussian Decoder**: ∼22 hours (100K iters on 32×H20, using frozen diffusion latents)
- **Enhanced Diffusion**: ∼59 hours (50K iters on 32×H20, initialized from 4D-Aware Diffusion checkpoint)

## A.3 GEOMETRIC CONSISTENCY AND MULTI-VIEW COHERENCE EVALUATION

To address concerns about geometric consistency and multi-view coherence, we evaluate our method using novel-view synthesis metrics following OmniScene's (Wei et al., 2024) protocol. We compare against feed-forward 3D reconstruction methods on the nuScenes validation set, measuring PSNR, SSIM, and LPIPS to assess the geometric fidelity and multi-view consistency of the generated scenes.

Table 6: Geometric consistency and multi-view coherence evaluation. We follow OmniScene's protocol for novel-view synthesis evaluation.

| Model | PSNR ↑ | SSIM ↑ | LPIPS ↓ |
|---|---|---|---|
| PixelSplat (Charatan et al., 2024) | 21.51 | 0.616 | 0.372 |
| MVSplat (Chen et al., 2024b) | 21.61 | 0.658 | 0.295 |
| OmniScene (Wei et al., 2024) | 24.27 | 0.736 | 0.237 |
| **Ours** | **24.59** | **0.912** | **0.147** |

As shown in Tab. 6, our method achieves superior performance across all metrics, demonstrating strong geometric consistency and multi-view coherence. The significantly higher SSIM (0.912 vs. 0.736) and lower LPIPS (0.147 vs. 0.237) compared to OmniScene indicate that our 4D Gaussian

representation effectively maintains structural coherence across different viewpoints. These results confirm that our approach not only generates high-quality videos (as measured by FID/FVD in the main paper) but also preserves geometric consistency and multi-view coherence essential for autonomous driving applications.

## B    INFERENCE SPEED COMPARISON

In Tab. 7, we compare the inference speed of our pipeline with MagicDrive-V2 (Gao et al., 2025) and Cosmos-transfer1 (Alhaija et al., 2025) on a single NVIDIA H20 GPU, producing a 17-frame, 6-view video at 424×800 resolution. Although our model employs two diffusion modules, the use of rectified flow with only 8 sampling steps keeps the inference speed comparable to others, which typically require over 30 steps.

Table 7: Efficiency comparison on novel scene generation. We report runtime breakdown and GPU memory usage for different methods.

| Method | Resolution | Diff-1 (s) | Gs Dec. (s) | Diff-2 (s) | VAE Dec. (s) | Total (min) | Device | GPU Mem (GB) |
|---|---|---|---|---|---|---|---|---|
| MagicDrive-V2 | 424×800 | 215.35 | - | - | 15.83 | 3.85 | H20 | 26 |
| Cosmos-transfer1 | 424×800 | 126.37 | - | - | 4.14 | 2.18 | H20 | 17 |
| Ours | 424×800 | 66.56 | 0.84 | 66.35 | 16.10 | 2.50 | H20 | 22 |

## C    MORE VISUALIZATION RESULTS

To better illustrate our Gaussian representation, we provide visualizations in Figs. 7, which demonstrate its high structural fidelity.

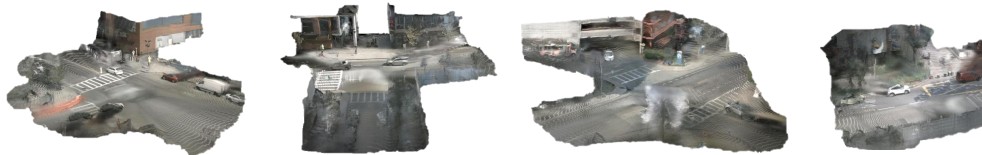

Figure 7: Visualizations of our Gaussians representation.

Further, our method produces fully controllable videos without relying on any reference frames, while simultaneously supporting high-quality novel-view synthesis. Figs. 8–13 showcase qualitative results of multi-trajectory generation without RGB inputs, clearly demonstrating the effectiveness and robustness of our model.

We include a series of generated novel-view videos in the supplementary material to further validate the quality of our results. Specifically, the supplementary videos correspond to two novel trajectories parallel to the original path, shifted by ±2 m to the left and right.

## D    THE USE OF LARGE LANGUAGE MODELS (LLMS)

A large language model (ChatGPT) was only used after completing the draft to polish wording and improve readability. It did not contribute to research or content creation. The authors take full responsibility for the final text.

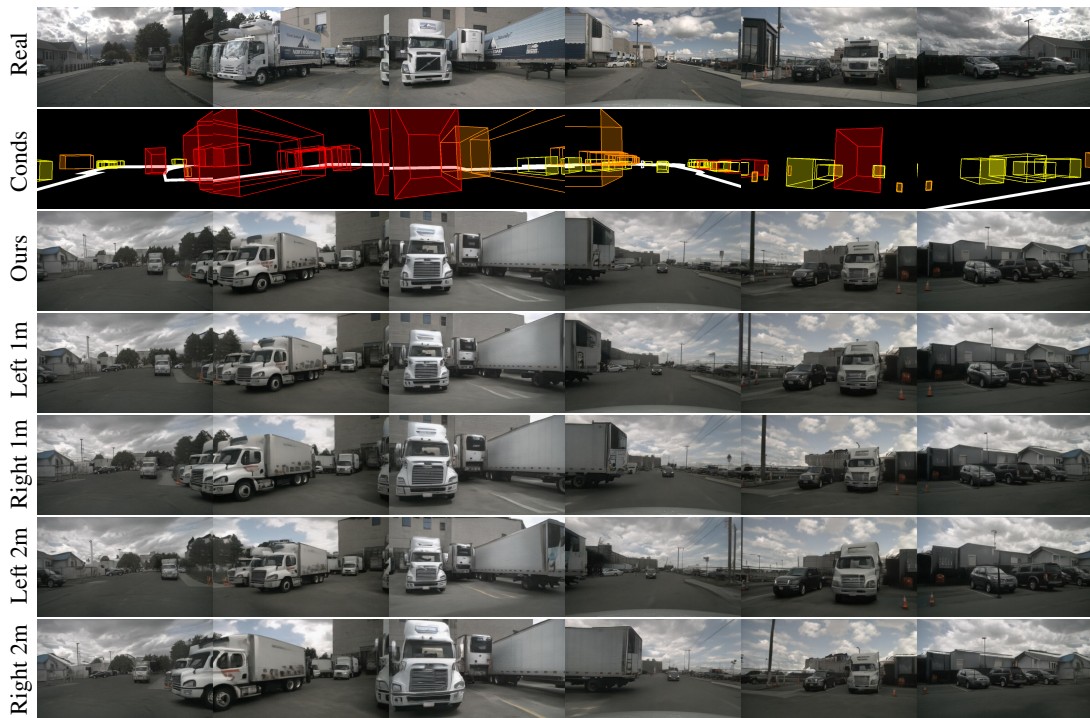

Figure 8: Multi-trajectory generation (from center to ±1m / ±2m lateral shifts) without RGB inputs.

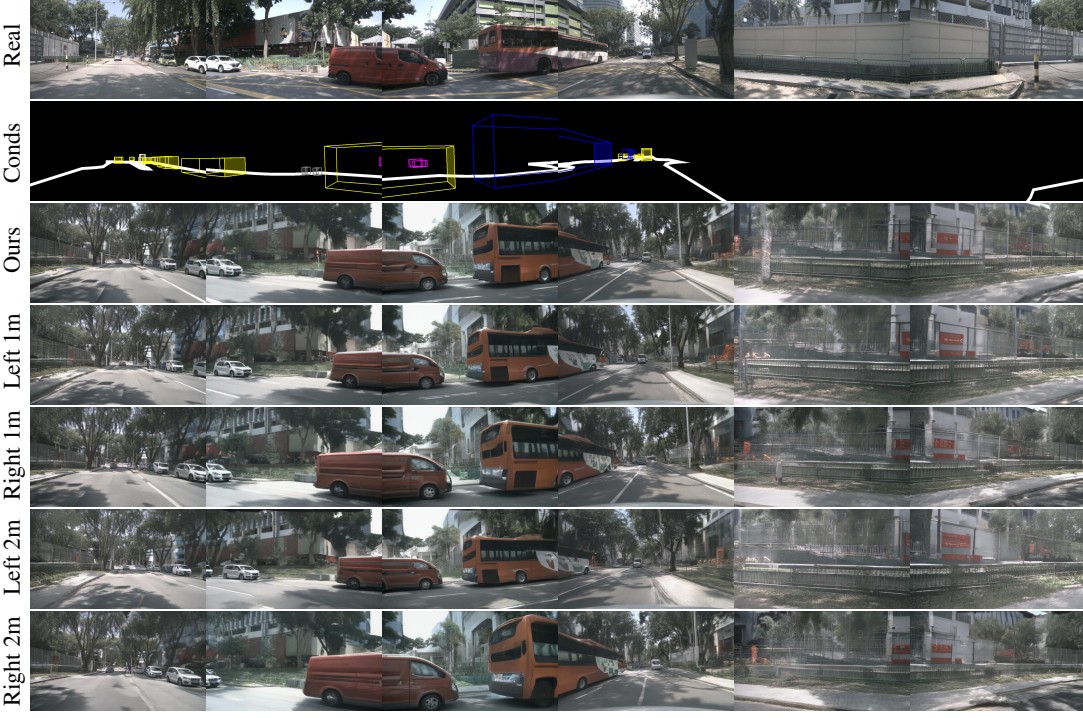

Figure 9: Multi-trajectory generation (from center to ±1m / ±2m lateral shifts) without RGB inputs.

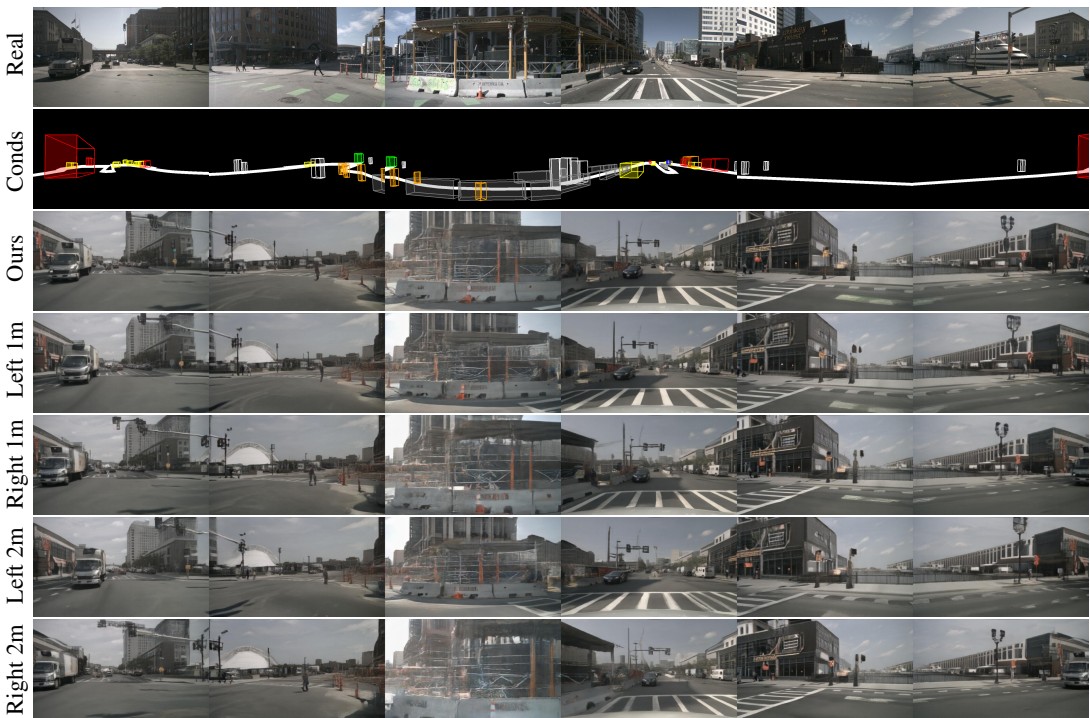

Figure 10: Multi-trajectory generation (from center to ±1m / ±2m lateral shifts) without RGB inputs.

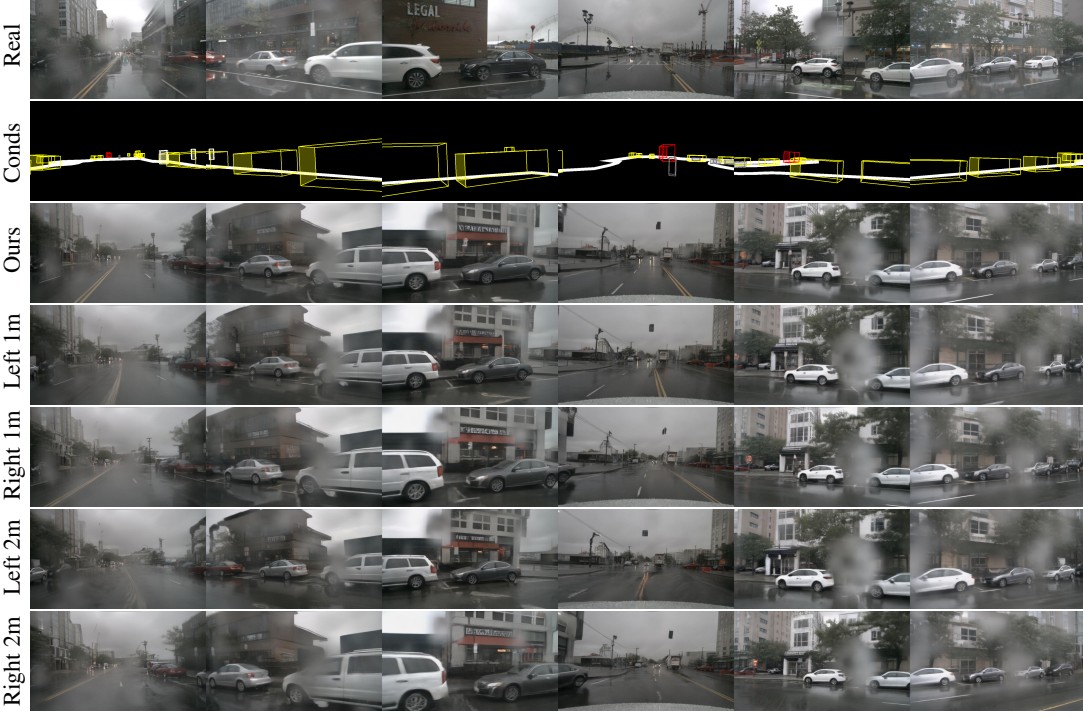

Figure 11: Multi-trajectory generation (from center to ±1m / ±2m lateral shifts) without RGB inputs.

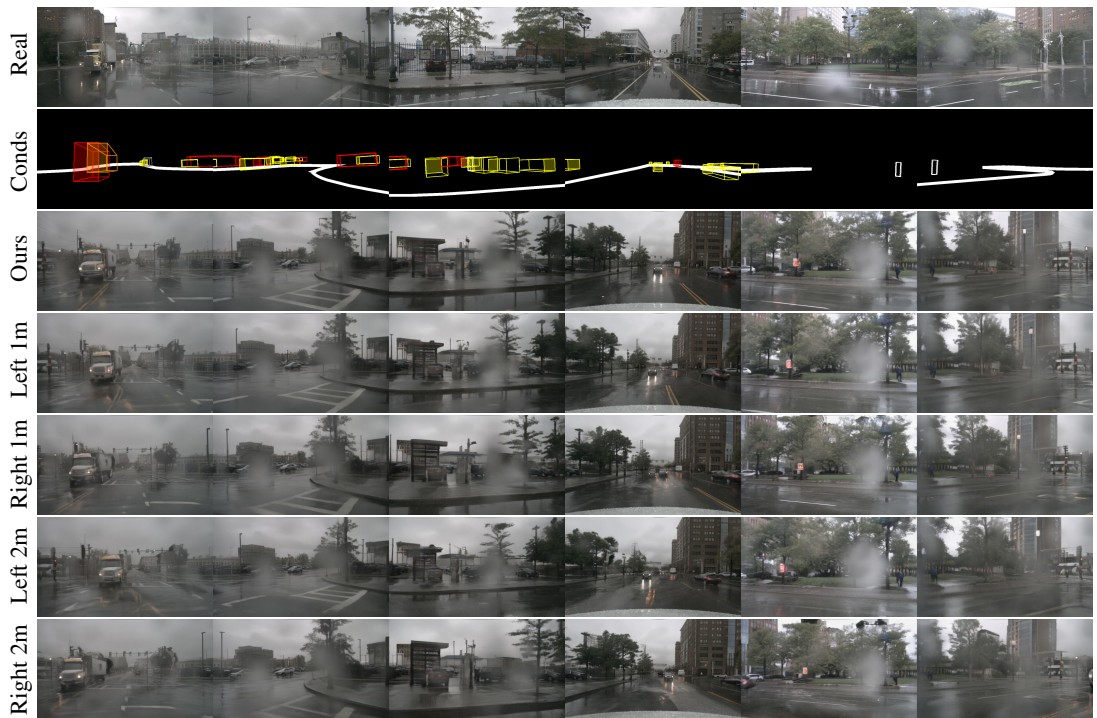

Figure 12: Multi-trajectory generation (from center to ±1m / ±2m lateral shifts) without RGB inputs.

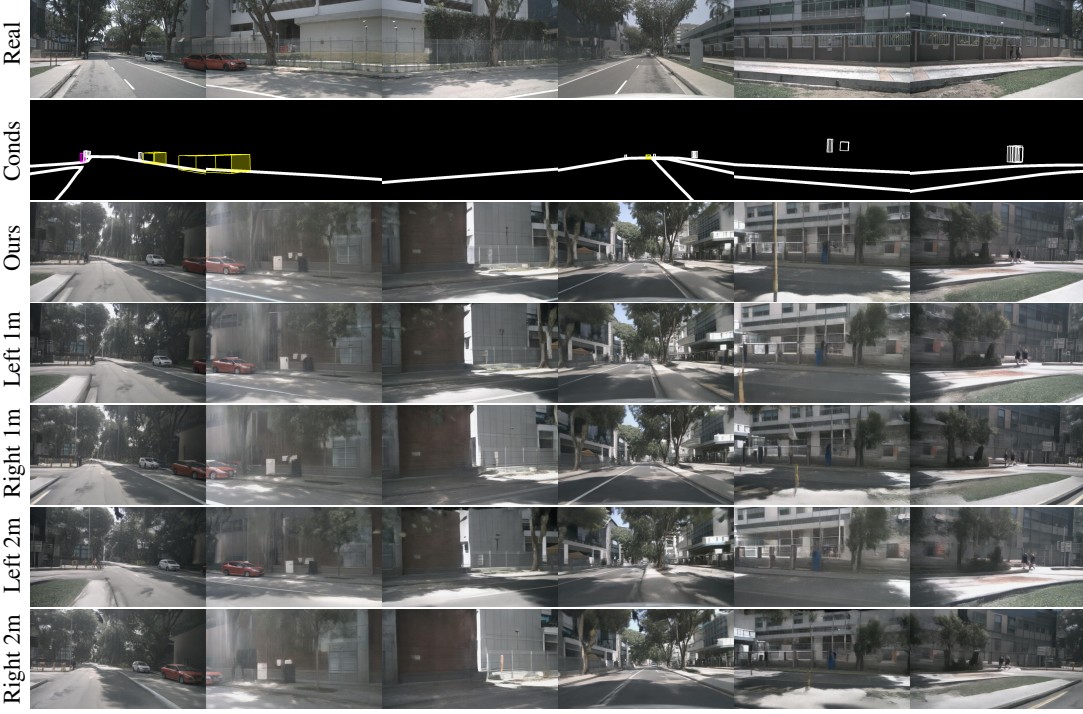

Figure 13: Multi-trajectory generation (from center to ±1m / ±2m lateral shifts) without RGB inputs.

