# OpenReview forum: "WorldSplat: Gaussian-Centric Feed-Forward 4D Scene Generation for Autonomous Driving"
_ICLR.cc/2026/Conference — ICLR 2026 Poster_

### Official Review · Reviewer_HFQu · 2025-10-22

**Soundness:** 2
**Presentation:** 3
**Contribution:** 2
**Rating:** 4
**Confidence:** 4

**Summary:**

This paper addresses novel-view synthesis in driving-scene generation and introduces WorldSplat, a feed-forward 4D generation framework bridging video synthesis and 3D/4D reconstruction. Leveraging a 4D-aware latent diffusion model to produce pixel-aligned Gaussians, followed by refined video diffusion, it generates high-fidelity, temporally and spatially consistent multi-track novel-view videos. Experiments on benchmark datasets show its effectiveness in producing realistic and controllable driving scenes.

**Strengths:**

1. The paper is well written, and the figures are clear and informative.
2. The method demonstrates strong performance in generation metrics.
3. The motivation is reasonable — leveraging Gaussian splatting to ensure geometric consistency, while incorporating diffusion models to enhance quality on novel regions.

**Weaknesses:**

1. The main issue with the paper lies in the inconsistency between the motivation and the experiments metrics. The method is only evaluated using FID and FVD, which primarily measure generation quality. These metrics fail to capture geometric consistency, as well as multi-view and temporal coherence, all of which are missing from the validation. When comparing with NVS methods, it is usually also report reconstruction metrics (psnr, ssim) on the original viewpoints to ensure consistency. FID and FVD alone only reflect perceptual generation quality, while many details may differ significantly — as can be clearly observed in the appendix figures, where the generated backgrounds differ noticeably from the original real images.
2. I have some concerns about the proposed method. It seems that the success of novel view rendering may largely stem from the strong conditioning provided. If the projected boxes and trajectories are given, the overall object layout and scene distribution are already well constrained. Previous generative methods could also leverage this conditioning to perform novel view generation. The real challenge lies in maintaining background consistency, such as the mismatch of elements like utility poles observed in the appendix visualizations. Therefore, it remains unclear how much geometric consistency is truly contributed by the Gaussian-centric module, and how much is simply due to the strong conditioning.
3. Since the method is based on 4D Gaussian Splatting, it would be important to include experiments demonstrating temporal consistency. Qualitative visualizations could also help illustrate this aspect. Currently, the appendix only provides static rendering results, which are insufficient to evaluate the model’s performance on temporal coherence.
4. Table 4(a) lacks a comparison with the BEVFormer baseline.

**Questions:**

1. With the rapid development of generative models, stronger models naturally lead to better generation performance. Compared with previous methods, is the performance improvement mainly due to the proposed approach or the use of a more powerful diffusion model?
2. Are Diff1, the GS decoder, and Diff2 trained jointly, or are they optimized in separate stages? It’s not entirely clear from the Training Details section.

---

> ### Author Response · Authors · 2025-11-21
> **Response to Reviewer HFQu**
>
> Dear Reviewer HFQu,
>
> We sincerely appreciate your detailed and constructive review regarding evaluation metrics and experimental design. We acknowledge that our unclear descriptions may have caused confusion. Specifically, we want to clarify that **the noticeable differences in generated backgrounds compared to original real images are intentional and demonstrate our method's controllable generation capability without RGB input.** Below we provide detailed clarifications and will incorporate these improvements in the final version.
>
> ---
>
> **[Weakness 1] Metrics only measure generation quality (FID/FVD) but fail to capture geometric consistency, multi-view coherence, and temporal consistency**
>
> **A1:**
> We have evaluated geometric consistency and multi-view coherence using novel-view synthesis metrics following OmniScene's protocol:
>
> | Model                   | PSNR ↑   | SSIM ↑   | LPIPS ↓  |
> |-------------------------|--------|--------|--------|
> | PixelSplat              | 21.51  | 0.616  | 0.372  |
> | MVSplat                 | 21.61  | 0.658  | 0.295  |
> | OmniScene               | 24.27  | 0.736  | 0.237  |
> | **Ours** | **24.59** | **0.912** | **0.147** |
>
> Our method achieves the best performance across all metrics, directly measuring structural consistency and perceptual quality.
> Regarding background variations in appendix figures, we clarify this in **Weakness 2** below.
>
>
> **[Weakness 2] Unclear how much geometric consistency comes from Gaussian-centric module vs. strong conditioning**
>
> **A2:**
> The examples in our appendix (Figures 8-13) intentionally demonstrate **pure generation without RGB input**—using only conditions (layout, boxes, trajectories, and captions) to showcase our model's generative capability.
>
> Importantly, these visualizations reveal that our generated novel views (from center to ±1m, ±2m lateral shifts) **maintain consistent foreground and background geometry among themselves**, which is only possible through our explicit 4D Gaussian representation. Simple conditioning alone cannot enforce such inter-view geometric consistency without an explicit 3D structure.
>
> Our model also supports real image input for novel-view video generation. To further demonstrate geometric consistency when RGB input is available, we have added real image examples in **Figure 5 of the revision**, showing our model's ability to maintain background consistency with real images.
>
>
>
> **[Weakness 3] Missing temporal consistency experiments and visualizations**
>
> **A3:**
> We have included a series of generated novel-view videos in the supplementary material to validate the temporal consistency of our results. Specifically, the supplementary videos show novel trajectories parallel to the original path, shifted by ±2m laterally, demonstrating smooth temporal transitions and consistent geometry across frames.
>
>
> **[Weakness 4] Table 4(a) lacks BEVFormer baseline comparison**
>
> **A4:**
> Here is the updated Table 4(a) with BEVFormer baseline comparison:
>
> **Table 4(a): Domain gap validation of generated data on driving perception with pretrained BEVFormer**
>
> | Method | mIoU↑ | mAP↑ |
> |--------|-------|------|
> | MagicDrive  | 18.34 | 11.86 |
> | MagicDrive3D  | 18.27 | 12.05 |
> | MagicDrive-v2  | 22.40 | 18.17 |
> | DiVE  | 35.96 | 24.55 |
> | **Ours** | **38.49** | **29.34** |
> | **Real** | **44.09** | **34.51** |
>
> Our method achieves the best performance among all generation methods (**38.49 mIoU, 29.34 mAP**), outperforming DiVE by **+2.53 mIoU** and **+4.79 mAP**. Notably, our generated data reaches **87.3%** of real data performance, demonstrating strong domain alignment with actual driving scenarios for perception model.
>
>
> **[Question 1] Is performance improvement mainly from proposed approach or more powerful diffusion model?**
>
> **A5:**
> Please refer to **Section 1 of the General Response** for our controlled comparison with identical backbones. With the same OpenSora v1.2 backbone, WorldSplat outperforms DiVE by **20.47 FVD** (-21.6%), demonstrating clear gains from our Gaussian-centric 4D generation approach rather than from a more powerful diffusion backbone.
>
> **[Question 2] Are Diff1, GS decoder, and Diff2 trained jointly or in separate stages?**
>
> **A6:**
> As mentioned in Section 3.1 of our main paper, the 4D-Aware Diffusion model, Gaussian Decoder, and Enhanced Diffusion model **are trained independently** in a staged manner and integrated together during inference. This staged training approach ensures stable optimization and allows each component to specialize in its respective task.

---

> > ### Comment · Reviewer_HFQu · 2025-11-25
> >
> > Thank you for the response and the additional experiments. The reconstruction metrics indeed strengthen the evidence for geometric consistency. However, I would like to clarify a few points regarding the response.
> >
> > First, novel view generation is typically defined as generating new viewpoints within the same underlying scene, rather than unconstrained free-form generation. If the intention is to showcase pure generative capability, I recommend adjusting the caption accordingly.
> >
> > Second, I do not fully agree with the statement “only possible through our explicit 4D Gaussian representation.” Methods such as FreeVS and StreetCrafter have demonstrated strong consistency under color–point-cloud conditioning, suggesting that such capability is not unique to 4D Gaussian representations.
> >
> > Third, regarding Weakness 3: similar video effects can be achieved by generating a novel viewpoint from the first frame and then maintaining consistency through standard video generation; this alone does not convincingly highlight the benefits of 4D Gaussian modeling. Moreover, the presented camera trajectories appear to be simple translations. It would be more convincing to include results along arbitrary trajectories—e.g., combining left and right shifts or slightly more complex motion paths.

---

> > > ### Author Response · Authors · 2025-11-25
> > > **Response to Reviewer HFQu**
> > >
> > > Dear Reviewer HFQu, we sincerely thank you for the valuable feedback on our response. We would like to provide further clarifications and appreciate your continued attention.
> > >
> > > **[Q1] Clarification on the definition of generating new viewpoints**
> > >
> > > Thank you for the helpful reminder. We agree that novel-view synthesis typically refers to generating new camera viewpoints of the same underlying scene. To avoid conflation, we have revised the captions of Appendix Figures 8-13 from “Novel View Generation” to  “Multi-trajectory generation (from center to ±1m / ±2m lateral shifts) without RGB inputs,”  which more accurately reflects the intended meaning.
> > >
> > > **[Q2] Regarding the statement about 4D Gaussian representation**
> > >
> > > Thank you for pointing out the two related great works. We acknowledge that explicit 4D Gaussian modeling is not the only way to achieve geometric consistency. Prior works such as FreeVS and StreetCrafter indeed show strong consistency under color-point-cloud conditioning.
> > > At the same time, our 4D Gaussian representation still offers two practical advantages compared with these approaches:
> > >
> > > (1) FreeVS and StreetCrafter require LiDAR to obtain color-point-cloud conditioning, while our method **does not rely on LiDAR**, making it naturally compatible with pure-vision autonomous driving systems such as Tesla-style pipelines.
> > >
> > > (2) LiDAR point clouds can sometimes be too sparse to maintain consistency when synthesizing larger viewpoint shifts.  In contrast, our 4D Gaussians represent the scene as soft, continuous volumetric ellipsoids, providing denser and more stable geometric support for novel-view generation. This advantage is reflected in the results of Table 2 in our main paper. For convenience, we show the comparison below:
> > > | Method | Shift ±1m FID ↓ | Shift ±1m FVD ↓ | Shift ±2m FID ↓ | Shift ±2m FVD ↓ | Shift ±4m FID ↓ | Shift ±4m FVD ↓ |
> > > |--------|------------------|------------------|------------------|------------------|------------------|------------------|
> > > | FreeVS | 51.26 | 431.99 | 62.04 | 497.37 | 77.14 | 556.14 |
> > > | **Ours** | **8.25** | **40.17** | **11.26** | **47.41** | **13.38** | **64.07** |
> > >
> > > **[Q3] Do not convincingly highlight the benefits of 4D Gaussian modeling**
> > >
> > > We agree that prior works such as FreeVS and StreetCrafter can also generate novel-view videos. As discussed in **Q2**, within a LiDAR-free framework, our 4D Gaussian modeling provides a spatial representation with denser geometry than the LiDAR-conditioned representations used in FreeVS and StreetCrafter. Table 2 in our main paper further demonstrates the superior performance of our method. Following your suggestion, we will additionally include more diverse novel-view visualizations under customized and more complex ego trajectories to better illustrate the model’s capability.

---

> > > > ### Comment · Reviewer_HFQu · 2025-11-28
> > > >
> > > > Thanks for the further response to my comments.
> > > >
> > > > To strengthen the paper, I suggest: 1) more clear demonstrations of novel view synthesis results; 2) an in-depth discussion comparing with previous methods and elaborating on the advantages of the 4D Gaussian representation; 3) additional results of more diverse trajectory generation (visual demos) to verify the method’s practical value.
> > > >
> > > > All my previous concerns have been fully addressed, and I am glad to raise my final score to 6.

---

> > > > > ### Author Response · Authors · 2025-11-28
> > > > > **Response to Reviewer HFQu**
> > > > >
> > > > > We sincerely thank the reviewer for the positive update and for raising the final score.
> > > > > We appreciate the constructive suggestions and will incorporate these improvements in the revision of the paper.
> > > > > Thank you again for the thoughtful and encouraging feedback.

---

### Official Review · Reviewer_9j61 · 2025-10-27

**Soundness:** 3
**Presentation:** 3
**Contribution:** 3
**Rating:** 6
**Confidence:** 4

**Summary:**

In this paper, the authors propose a novel feed-forward framework for 4D driving-scene generation, which can integrate multi-modal information via a 4D-aware latent diffusion model to produce pixel-aligned 4D Gaussians in a feed-forward manner. What's more, the authors use an enhanced video diffusion model to refine the novel view videos rendered from these Gaussians. The experiments show that WorldSplat can generate high-fidelity novel view driving videos.

**Strengths:**

1. The authors focus on the limitations of existing methods—2D video generation lacks 3D consistency, while 3D/4D reconstruction lacks generative capabilities—and select 4D Gaussians as the core representation to improve 3D consistency, bridging the gap between generation and reconstruction.
2. The framework can generate a dynamic 4D Gaussian representation via a latent Gaussian decoder and then render novel-view videos through these Gaussians. This design effectively enhances both spatial and temporal geometric consistency, addressing the 3D inconsistency issue of existing 2D video generation methods.
3. Experiments on the nuScenes dataset show that the rendered novel-view videos have high fidelity(Tab.2) and the framework exhibits strong generative capabilities(Tab.1)

**Weaknesses:**

1. It seems that the training of the Gaussian Decoder relies on the seg loss, which requires additional time consumption due to the external use of SegFormer. Could the authors discuss the total training time of the framework, particularly the time overhead introduced by SegFormer?

**Questions:**

1. I am confused about Fig. 2. It seems that only the 4D-Aware Diffusion Model needs to be trained, but in the paper, the Latent Gaussian Decoder also needs to be trained.
2. It is hoped that the authors can provide more descriptions regarding the training time of the entire framework and all its training stages.

---

> ### Author Response · Authors · 2025-11-21
> **Response to Reviewer 9j61**
>
> Dear Reviewer 9j61,
>
> We sincerely thank you for the positive evaluation and recognition of our approach bridging generation and reconstruction with 4D Gaussians. We address each concern below and will incorporate the suggested clarifications and improvements in the final version.
>
> ---
>
> **[Weakness 1] Training time of the framework, particularly the overhead from SegFormer**
>
> **A1:**
> Thank you for raising this important concern. We clarify that **SegFormer is only used during data preprocessing, not during model training**, introducing zero training-time overhead. Specifically, SegFormer-B5 pre-computes semantic segmentation masks for all nuScenes images using 8×H20 GPUs in approximately 9.5 hours (one-time cost). During training, our Gaussian Decoder predicts segmentation masks, and the segmentation loss directly compares these predictions against the pre-computed ground truth masks, requiring no additional SegFormer inference.
>
> **Complete Training Time Breakdown:**
>
> As detailed in **Appendix A.2** of our paper, the training is organized into four sequential stages on 32× H20 GPUs:
>
> | Stage | Description | Steps* | Resolution (6 views) | Time |
> |-------|-------------|-----------|----------------------|------|
> | **Stage 1** | Fine-tune from OpenSora v1.2 with layout/sketch control | 60K | 256×256 | ~32h |
> | **Stage 2** | Mixed-resolution training with varying frame lengths | 40K | 144p–360p | ~28h |
> | **Stage 3** | Switch to rectified flow at low resolution | 20K | 144p–360p | ~14h |
> | **Stage 4** | High-resolution fine-tuning with rectified flow | 60K | 480p–full | ~83h |
> | **Total** | **4D-Aware Diffusion Training** | **180K** | - | **~157h** |
>
> *Steps are reported for 8-GPU training. For 32-GPU training, divide by 4 (e.g., Stage 1: 15K steps).
>
> **Additional Training Stages:**
>
> - **Gaussian Decoder**: ~22 hours (100K iters on 32×H20, using frozen diffusion latents)
> - **Enhanced Diffusion**: ~59 hours (50K iters on 32×H20, initialized from 4D-Aware Diffusion checkpoint)
>
>
> **[Question 1] Figure 2 is confusing – it seems only the 4D-Aware Diffusion needs training, but the Latent Gaussian Decoder also needs training**
>
> **A2:**
> As mentioned in Section 3.1 of our main paper, **all three modules require training**: the 4D-Aware Diffusion model, Gaussian Decoder, and Enhanced Diffusion model are trained independently in a staged manner and integrated together during inference. Detailed training procedures for all three stages are provided in our **Weakness 1** response above.
>
> **[Question 2] Need more descriptions regarding training time of entire framework and all training stages**
>
> **A3:**
> Thank you for this request. Please refer to **Weakness 1** above, where we provide comprehensive details on the training time for the entire framework across all training stages, including stage-by-stage breakdowns, iteration counts, resolutions, and GPU-hour requirements. The complete training takes approximately **238 hours (~10 days) on 32×H20 GPUs** (157h for 4D-Aware Diffusion + 22h for Gaussian Decoder + 59h for Enhanced Diffusion), plus a one-time 9.5-hour preprocessing step for SegFormer.

---

> > ### Comment · Reviewer_9j61 · 2025-11-24
> >
> > Thank you for your reply. It has addressed my concerns. I’d like to retain my original score.

---

> > > ### Author Response · Authors · 2025-11-25
> > > **Response to Reviewer 9j61**
> > >
> > > We sincerely thank the reviewer for the thoughtful evaluation and for maintaining the rating. We truly appreciate the constructive feedback, which has helped us further improve the clarity and completeness of the paper.

---

### Official Review · Reviewer_t1xN · 2025-10-28

**Soundness:** 3
**Presentation:** 3
**Contribution:** 3
**Rating:** 6
**Confidence:** 4

**Summary:**

This paper proposes a Gaussian-centric, feed-forward 4D scene generation method for autonomous driving, called WorldSplat. The work explores how to combine generative models with accurate 4D reconstruction. WorldSplat introduces three key modules: the 4D-aware latent diffusion module, which transforms user-defined control conditions into temporally coherent latents; the latent Gaussian decoder, which converts these latents into explicit 4D Gaussian scene geometry; and the enhanced diffusion refinement model, which further improves the quality of rendered frames. Compared to other methods, WorldSplat achieves state-of-the-art performance.

**Strengths:**

- WorldSplat elegantly combines the generative ability of novel scene generation with the custom scene reconstruction capability of Gaussian scene (GS) models.
- WorldSplat achieves state-of-the-art performance under various conditions, including both custom and novel-view synthesis, outperforming other leading methods.

**Weaknesses:**

- The ablation study in this work is somewhat simplistic, and the authors tend to overstate their claims.
- The novelty of the module design is limited, and the methods section delves into well-known concepts, such as the diffusion process, without offering significant new insights.

**Questions:**

- In L226, it says "our decoder supports over 48 simultaneous input views." Could you clarify how the model supports 48 views? Why can't prior feed-forward reconstruction models achieve this capacity?
- Both the introduction and the ablation study mention the "Mixed Aug" technology, but I don't believe this technology is a key contribution of the current work. Could you clarify why it is mentioned so frequently?
- In WorldSplat, I would like to know how the performance is affected without the reprojection conditions in the Enhanced Diffusion Model. It seems that the restricted views sampled from the Aggregated 4D Gaussians are somewhat aligned with the conditions. Could this lead to performance degradation in novel-view synthesis?

---

> ### Author Response · Authors · 2025-11-21
> **Response to Reviewer t1xN**
>
> Dear Reviewer t1xN,
>
> We sincerely thank you for recognizing our elegant combination of generation and reconstruction capabilities and the state-of-the-art performance. We address each concern below and will incorporate the suggested clarifications and improvements in the final version.
>
> ---
>
> **[Weakness 1] Ablation study is somewhat simplistic, and authors tend to overstate their claims**
>
> **A1:**
> Please refer to **Section 2 of the General Response** for our expanded ablation study with detailed component-wise analysis, including the new Version E that specifically addresses your Question 3 regarding reprojection conditions.
>
> **[Weakness 2] Novelty of module design is limited; methods section discusses well-known concepts**
>
> **A2:**
> Our key contribution lies precisely in providing a new perspective to solve this fundamental dilemma: existing generation methods lack 3D consistency for novel-view synthesis, while reconstruction methods inherently lack generative capabilities. WorldSplat addresses this through a Gaussian-centric feed-forward framework that unifies both capabilities—enabling controllable scene generation while maintaining geometric consistency for high-quality novel-view synthesis. This integration of 4D-aware latent diffusion, feed-forward Gaussian decoder, and enhanced video refinement has not been explored before in autonomous driving generation. Beyond technical novelty, we will open-source our code to enable the community to build upon this work, contributing a practical and reproducible solution to advance research in this direction.
>
> **[Question 1] How does the decoder support 48 simultaneous input views? Why can't prior feed-forward reconstruction models achieve this?**
>
> **A3:**
> Our decoder's 48-view capacity stems from operating in **compressed VAE latent space** with **efficient flash-attention mechanisms**. Specifically, our decoder processes VAE-compressed latents with 8× spatial downsampling and 4 channels rather than raw images, providing substantial memory reduction compared to pixel-space processing. Cross-view and temporal attention operates on these compact latents using flash-attention, which further reduces memory consumption and avoids expensive operations, enabling processing of many more views within the same memory budget.
>
> However, prior models typically cannot achieve this capacity.
> For example, PixelSplat is built around a stereo-based depth lifting pipeline tailored to two-view geometry, making it difficult to extend efficiently to many more views.
> MVSplat performs geometry-aware fusion across views via per-view cost-volume computation and cross-view attention, but its computational cost grows linearly with the number of input cameras, making large-view settings impractical.
> OmniScene is specifically designed for single-frame 6-camera settings without temporal modeling, limiting its applicability to long sequences or larger view counts.
>
> **[Question 2] Why "Mixed Augs" is mentioned frequently?**
>
> **A4:**
> We acknowledge that Mixed Augmentation is not a key contribution of our work. We mentioned it in Section 3.4 and the ablation study to provide readers with a comprehensive understanding of our model's implementation details. As shown in our ablation study (Version C → F), Mixed Augmentation contributes modestly to the final performance (FVD: -3.32, FID: -0.34). It serves as a practical design choice to improve the enhanced diffusion model's robustness by bridging the quality gap between source and novel views during training, rather than a core technical novelty. We will clarify this in the revision to avoid giving the impression of overstating its importance.
>
> **[Question 3] How is performance affected without reprojection conditions in the Enhanced Diffusion Model?**
>
> **A5:**
> Please refer to **Section 2 of the General Response**, specifically the analysis of Version E (without reprojection conditions) vs. Version F (full model). Removing reprojection conditions leads to 8.9% increase in FVD and 7.2% increase in FID, demonstrating its contribution to enforcing 3D geometric constraints across views.

---

### Official Review · Reviewer_VA3W · 2025-10-29

**Soundness:** 3
**Presentation:** 4
**Contribution:** 3
**Rating:** 6
**Confidence:** 4

**Summary:**

Current Driving scene generation methods lack 3D consistency for novel-view synthesis, while reconstruction methods cannot generate unseen scenarios. This paper introduces WorldSplat, a feed-forward framework that bridges this gap using explicit 4D Gaussian representations. The approach employs: (1) a 4D-aware latent diffusion model to generate multi-modal latents (RGB, depth, semantic) from control conditions, (2) a Gaussian decoder that produces pixel-aligned 4D Gaussians with static-dynamic decomposition for efficient novel-view rendering, and (3) an enhanced diffusion model for refinement. Experiments on nuScenes demonstrate impressive performance in video generation and novel-view synthesis, with improvements on downstream perception tasks.

**Strengths:**

- The submission contributes a novel feed-forward Gaussian decoder which effectively uses the latents from the 4D latent diffusion model to create accurate static/dynamic decomposed 4D Gaussians.
- The method effectively addresses a key limitation in the field by bridging the gap between video generation methods (which may contain artifacts and inconsistencies) and reconstruction methods (which lack generative capability). The use of explicit 4D Gaussian representations as an intermediate representation is clever and well-justified.
- The paper demonstrates significant improvements across multiple metrics, including FVD/FID scores for video generation (Table 1), novel-view synthesis quality (Table 2), and  downstream driving tasks (Table 4)
- The paper is well structured and lays out the rationale for the architecture clearly. The combination of a generative pipeline to create 4D Gaussians paired with a final diffusion model for inpainting unseen regions and overall refinement is well motivated by related work.

**Weaknesses:**

W1. While the improvements in FID and FVD compared to baselines even at varying degrees of camera displacements is solid, the work focuses on this improvement without examining or ablating how much of this improvement can be attributed to the backbone model (OpenSora v1.2) versus their own models.The downstream task comparisons also show meaningful improvements in mIoU and mAP for BEVFormer. Of note is the fact that the closest model in terms of performance to WorldSplat is DiVE, which also employs an OpenSora backbone in contrast to the MagicDrive works. The paper is missing a comparison of FID and FVD against DiVE despite it being a comparable model in terms of the end video diffusion model used to generate RGB frames.

W2. While the paper shows successful examples, it lacks in-depth discussion of failure cases and ablations. Table 3 demonstrates the improvements when using all of WorldSplat's individual components, but doesn't provide any explanation or intuition behind the results. For example, Version D shows that without enhanced diffusion, FVD degrades from 47.41 to 107.58 (>2x worse), suggesting the end diffusion model is doing a lot of the heavy lifting to uphold the FID and FVD scores rather than the generated Gaussians (despite the consistently better performance with displacement compared to other methods). There's also little discussion of failure cases at all.

**Questions:**

Additional questions:

Q1. In reference to W1, can the authors provide an comparison between methods using modern video backbones like OpenSora for FID and FVD?

Q2. What amount of generated data was used in the mixed "Real + Ours" result in Table 4b?

Q3. How do you prevent redundant Gaussians from multiple views of the same static regions?

---

> ### Author Response · Authors · 2025-11-21
> **Response to Reviewer VA3W**
>
> Dear Reviewer VA3W,
>
> We sincerely thank you for the positive evaluation and recognition of our feed-forward Gaussian decoder design and comprehensive experimental validation. We address each concern below and will incorporate the suggested clarifications and improvements in the final version.
>
> ---
>
> **[Weakness 1] Backbone model contribution unclear – how much improvement comes from OpenSora vs. our method?**
>
> **A1:**
> Please refer to **Section 1 of the General Response** for our controlled comparison with identical backbones. With the same OpenSora v1.2 backbone, WorldSplat outperforms DiVE by **20.47 FVD** (-21.6%).
> This improvement stems from our explicit 4D Gaussian representation that enforces geometric coherence across viewpoints and temporal frames.
>
>
> **[Weakness 2] Lack of detailed ablation studies and failure case analysis**
>
> **A2:**
> Please refer to **Section 2 of the General Response** for our expanded ablation study with systematic component-wise validation.
>
> Additionally, we provide failure case analysis below:
>
> **Failure Case Analysis:**
>
> Our framework can fail when the customized ego trajectory enters completely unobserved regions (e.g., shifting a trajectory inside a building whose interior is never visible from original viewpoints). Without geometric priors from Gaussians' rendering, the enhanced diffusion model produces artifacts or blank regions. This limitation is inherent to view synthesis from limited observations and affects all reconstruction-based methods; future work could incorporate geometric inpainting or learned priors for heavily occluded regions.
>
> **[Question 1] Can authors provide comparison with modern video backbones like OpenSora for FID and FVD?**
>
> **A3:**
> Please refer to **Section 1 of the General Response**. We have added direct comparison with DiVE showing our method's contribution.
> We also add the backbone details in the Table 1 of the revision.
>
>
> **[Question 2] What amount of generated data was used in the mixed "Real + Ours" result in Table 4b?**
>
> **A4:**
> Following Panacea's setup, we use a **1:1 mixing ratio**: 28,130 real nuScenes samples + ~28,130 WorldSplat-generated samples. This achieves mAP **38.5** (+4.0) and NDS **50.1** (+3.2), outperforming Panacea's mAP **37.1** (+2.6) and NDS **49.2** (+2.3).
>
>
>
> **[Question 3] How do you prevent redundant Gaussians from multiple views of the same static regions?**
>
> **A5:**
> Redundancy is naturally avoided through our training design. We aggregate 3D Gaussians from all views into a unified 4D representation during training, then supervise with rendering losses across multiple viewpoints. This enforces the model to learn generating consistent, non-redundant Gaussians for static regions at inference time.

---

### Author Response · Authors · 2025-11-21
**General Response to All Reviewers**

We sincerely thank all reviewers and ACs for their constructive feedback and recognition of our work in bridging generation and reconstruction for autonomous driving scenes. Below we address two key concerns raised across multiple reviews.

## 1. Isolating Our Method's Contribution from Backbone Model

**Reviewers VA3W (W1) and HFQu (Q1) asked whether the improvements come from the backbone or our approach.**

We provide additional comparisons to isolate our method's contribution from the backbone model.

| Method | Backbone | FVD ↓ | FID ↓ |
|--------|----------|-------|-------|
| MagicDrive | SD-v1.5 | 218.12 | 16.20 |
| MagicDrive-v2 | CogVideoX | 94.84 | 20.91 |
| DiVE | OpenSora-v1.2 | 94.60 | unavaliable |
| **WorldSplat (Ours)** | **OpenSora-v1.2** | **74.13** | **8.78** |

With the **same backbone (OpenSora v1.2)**, WorldSplat outperforms DiVE by **20.47 FVD** (-21.6%) and achieves **8.78 FID**, demonstrating gains from our Gaussian-centric 4D approach, not from a stronger backbone.

## 2. Comprehensive Ablation Study Analysis

**Reviewers t1xN (W1, Q3) and VA3W (W2) requested detailed ablation analysis.**

Expanded ablation with systematic component validation:

| Version | 3D Gs | 4D Gs | Mixed Aug | Enhanced | C-Reproj | FVD ↓ | FID ↓ |
|---------|-------|-------|-----------|----------|----------|-------|-------|
| A       |       |       |           | ✓        | ✓        | 260.07 | 41.40 |
| B       | ✓     |       |           | ✓        | ✓        | 75.26  | 16.31 |
| C       | ✓     | ✓     |           | ✓        | ✓        | 50.73  | 11.60 |
| D       | ✓     | ✓     | ✓         |          | ✓        | 107.58 | 26.73 |
| **E (New)**      | **✓**     | **✓**     | **✓**        | **✓**       |          | **51.64**  | **12.07** |
| F | ✓ | ✓ | ✓ | ✓ | ✓    | 47.41 | 11.26 |

We systematically validate each component's contribution:

1. **3D Gaussians Representation (Version A → B)**:
Introducing 3D Gaussians as explicit scene representation significantly improves performance: FVD from 260.07 to 75.26 (-184.81), and FID from 41.40 to 16.31 (-25.09). This demonstrates that 3D Gaussians enable the model to generate spatiotemporally consistent videos with effective novel-view synthesis capability.

2. **4D Gaussian Aggregation (Version B → C)**:
Aggregating single-frame 3D Gaussians into a unified 4D representation further improves performance: FVD from 75.26 to 50.73 (-24.53), and FID from 16.31 to 11.60 (-4.71). Single-frame Gaussians are often too sparse, producing holes and aliasing in novel views. The 4D aggregation enhances spatiotemporal consistency by densifying the representation across time in dynamic scenes.

3. **Mixed Augmentation (Version C → F)**:
Mixed Augmentation contributes to the final improvement: FVD from 50.73 to 47.41 (-3.32), and FID from 11.60 to 11.26 (-0.34). As discussed in Line 292 of our paper, novel-view renderings at inference often appear inferior to source views. By degrading training source view quality through mixed augmentation, we improve the enhanced diffusion model's robustness and overall generation quality.

4. **Enhanced Diffusion (Version D → F)**:
The Enhanced Diffusion module provides substantial improvements: FVD drops dramatically from 107.58 to 47.41 (-60.17), and FID from 26.73 to 11.26 (-15.47). As detailed in Section 3.4 and illustrated in Figure 3 of our paper, this module addresses inherent limitations of Gaussian splatting: (1) low-quality renderings in unobserved regions, and (2) blurred reconstructions under strong ego motion without per-scene optimization. The enhanced diffusion model effectively refines these artifacts by inpainting missing content and sharpening fast-motion frames.

5. **Condition Reprojection (Version E → F)**:
Removing reprojection conditions leads to 8.9% increase in FVD (47.41 → 51.64) and 7.2% increase in FID (11.26 → 12.07). The rendering views from our 4D Gaussians already provide strong conditions for the enhanced diffusion model. The condition reprojection (Section 3.5 of our main paper) further enhances performance by enforcing explicit 3D geometric constraints across views and frames, though the improvement is moderate compared to other components since the 4D Gaussian renderings themselves already capture substantial geometric information.

## 3. Updates for Final Revision

We have incorporated the following improvements in the revision, with all updated content highlighted in blue:

1. Backbone comparison and DIVE performance in Table 1 (VA3W-W1, HFQu-Q1)
2. Expanded ablation with detailed analysis (t1xN-W1, VA3W-W2)
3. Geometric consistency metrics: PSNR/SSIM/LPIPS (HFQu-W1)
4. Training pipeline diagram with time breakdown (9j61-W1/Q2, HFQu-Q2)
5. Failure case discussion and limitations (VA3W-W2)
6. BEVFormer baseline in Table 4(a) (HFQu-W4)
7. New Figure 5 showing NVS with real RGB input for geometric consistency validation (HFQu-W2)

We thank all reviewers and ACs for their valuable feedback.

---

### Meta-Review · Area_Chair_gGj2 · 2025-12-30

**Summary:**

The reviewers’ concerns mainly include:
1.	The ablation studies have notable shortcomings, lacking well-designed experimental setups and thorough analysis, and missing necessary evaluation metrics.
2.	The novelty of the module design is limited.

**Reviewer Concerns:**

Most of the concerns have been addressed. However, some remain regarding more comprehensive experimental evaluation of the novel view synthesis results, a clearer and more detailed explanation of how the proposed method differs from prior approaches and how it benefits from the 4D Gaussian representation. The paper is also encouraged to present additional results demonstrating its performance under more challenging settings.

**Reviewer Scores:**

Reviewer HFQu tends to raise his score to 6, replacing the original score of 4 after the discussion. The other reviewers tend to maintain their original scores.

---

### Decision · Program_Chairs · 2026-01-26

Accept (Poster)